# Restoration of PITPNA in Type 2 diabetic human islets reverses pancreatic beta-cell dysfunction

Yu-Te Yeh[1,2,18], Chandan Sona[1,2,18], Xin Yan[3,4], Yunxiao Li[3], Adrija Pathak[5], Mark I. McDermott[6], Zhigang Xie[6], Liangwen Liu[7], Anoop Arunagiri [8], Yuting Wang[4], Amaury Cazenave-Gassiot [9,10], Adhideb Ghosh [11], Ferdinand von Meyenn [11], Sivarajan Kumarasamy[12,13], Sonia M. Najjar[12,13], Shiqi Jia [14], Markus R. Wenk[9,10], Alexis Traynor-Kaplan[15,16], Peter Arvan [8], Sebastian Barg [7], Vytas A. Bankaitis [5,6,17] & Matthew N. Poy [1,2,4] ✉

Defects in insulin processing and granule maturation are linked to pancreatic beta-cell failure during type 2 diabetes (T2D). Phosphatidylinositol transfer protein alpha (PITPNA) stimulates activity of phosphatidylinositol (PtdIns) 4-OH kinase to produce sufficient PtdIns-4-phosphate (PtdIns-4-P) in the trans-Golgi network to promote insulin granule maturation. *PITPNA* in beta-cells of T2D human subjects is markedly reduced suggesting its depletion accompanies beta-cell dysfunction. Conditional deletion of *Pitpna* in the beta-cells of *Ins*-Cre, *Pitpna*^*flox/flox* mice leads to hyperglycemia resulting from decreasing glucose-stimulated insulin secretion (GSIS) and reducing pancreatic beta-cell mass. Furthermore, *PITPNA* silencing in human islets confirms its role in PtdIns-4-P synthesis and leads to impaired insulin granule maturation and docking, GSIS, and proinsulin processing with evidence of ER stress. Restoration of *PITPNA* in islets of T2D human subjects reverses these beta-cell defects and identify *PITPNA* as a critical target linked to beta-cell failure in T2D.

Type 2 diabetes (T2D) is a non-autoimmune disease of impaired insulin signaling that afflicts ~10% of the population in the United State alone[1,2]. Both impaired insulin release and reduced beta-cell mass contribute to the beta-cell failure that occurs during T2D[3–5]. Beta-cell failure is calculated to associate with a 24–65% loss of measurable beta-cell mass and a 50–97% loss of secretory capacity after disease onset[6,7]. Prior to the events that cause this decline, beta-cells functionally accommodate peripheral insulin resistance for a limited time in two

[1]Johns Hopkins University, All Children's Hospital, St. Petersburg, FL 33701, USA. [2]Johns Hopkins University, Department of Medicine, Division of Endocrinology, Diabetes and Metabolism, Baltimore, MD 21287, USA. [3]Translational Neurodegeneration Section "Albrecht-Kossel", Department of Neurology, University Medical Center Rostock, Rostock 18147, Germany. [4]Max Delbrück Center for Molecular Medicine, Robert Rössle Strasse 10, Berlin 13125, Germany. [5]Department of Biochemistry & Biophysics, Texas A&M University, College Station, TX 77843, USA. [6]Department of Cell Biology & Genetics, Texas A&M Health Science Center, College Station, TX 77843, USA. [7]Medical Cell Biology, Uppsala University, 75123 Uppsala, Sweden. [8]Division of Metabolism, Endocrinology & Diabetes, University of Michigan Medical School, Ann Arbor, MI 48105, USA. [9]Singapore Lipidomics Incubator, Life Sciences Institute, National University of Singapore, 117456 Singapore, Singapore. [10]Department of Biochemistry and Precision Medicine TRP, Yong Loo Lin School of Medicine, National University of Singapore, 117597 Singapore, Singapore. [11]Laboratory of Nutrition and Metabolic Epigenetics, Department of Health Sciences and Technology, ETH Zurich, Schwerzenbach 8603, Switzerland. [12]Department of Biomedical Sciences, Heritage College of Osteopathic Medicine, Ohio University, Athens, OH 45701, USA. [13]Diabetes Institute, Heritage College of Osteopathic Medicine, Ohio University, Athens, OH 45701, USA. [14]The First Affiliated Hospital of Jinan University, Guangzhou, China. [15]Department of Medicine, University of Washington School of Medicine, Seattle, WA 98195, USA. [16]ATK Analytics, Innovation and Discovery, LLC, North Bend, WA 98045, USA. [17]Department of Chemistry, Texas A&M University, College Station, TX 77843, USA. [18]These authors contributed equally: Yu-Te Yeh, Chandan Sona. ✉e-mail: mpoy1@jhmi.edu

ways. First, beta-cells increase insulin production[2,4]. Second, beta-cells increase their proliferation to expand the pool of insulin-producing cells in order to compensate for increased metabolic demand[8–10]. However, beta-cells in T2D patients ultimately succumb to multiple complications that include endoplasmic reticulum (ER) stress, glucotoxicity, and dedifferentiation[11–14]. The precise mechanisms underlying the decline of both beta-cell secretion and mass remain unclear. As a result, worldwide efforts continue to focus on identifying the molecular bases of these defects[15–17].

Phosphoinositides define a set of chemically distinct phosphorylated derivatives of the glycerophospholipid phosphatidylinositol[18]. The central importance of phosphoinositide signaling in regulating cellular homeostasis in eukaryotes is demonstrated in two ways. First, the diversity of cellular activities regulated by phosphoinositide metabolism is striking. Phosphoinositide signaling controls cellular functions that range from membrane trafficking to receptor signaling at the plasma membrane, autophagy, transcription, mRNA transport, cytoskeleton dynamics, and numerous other activities[19–22]. Second, even subtle derangements in phosphoinositide metabolism contribute instrumentally to many diseases—including diabetes[23–25]. Phosphatidylinositol transfer proteins (PITPs) are highly conserved molecules that regulate the interface between lipid metabolism and cellular functions[26,27]. PITPs promote the activity of phosphatidylinositol (PtdIns) 4-OH kinases and PtdIns-4 phosphate (PtdIns-4-P) synthesis in eukaryotic cells[28–30]. There are at least three soluble PITPs expressed in mammals—PITPα/PITPNA, PITPβ/PITPNB, and rdgBβ/PITPnc1[31–33]. PITPNA and PITPNB share ~77% sequence identity, are encoded by distinct genes, and both PITPNA and PITPNB are characterized as transfer proteins for several phospholipids including PtdIns in vitro[34,35]. However, rather than functioning as inter-organelle lipid transfer proteins in cells, all available data are more consistent with PITPs serving as metabolic sensors that facilitate the presentation of PtdIns to PtdIns 4-OH kinases in vivo—thereby channeling PtdIns-4-P signaling to specific (and diverse) biological outcomes[36]. In that regard, PITPs contribute to secretory vesicle formation from the trans-Golgi network (TGN) to Ca$^{2+}$-activated secretion in permeabilized neuroendocrine cells, and to the regulation of Golgi dynamics in embryonic neural stem cells of the developing mouse neocortex[28,29,37–40].

Here we first demonstrate that functional ablation of *Pitpna* in murine beta-cells results in random-fed hyperglycemia due to both impaired glucose-stimulated insulin secretion (GSIS) and reduced beta-cell number. These defects are accompanied by induction of ER stress and deranged mitochondrial dynamics and performance. Consistent with the murine studies, we further show that expression of *PITPNA* (referred to as human *PITPNA* and mouse *Pitpna*) is markedly diminished in pancreatic islets of T2D human subjects compared to non-diabetic donors. Such a downregulation is of functional consequence as reduced *PITPNA* levels in isolated human islets compromised PtdIns-4-P synthesis in the Golgi system, impaired insulin granule maturation and docking, and induced both ER and mitochondrial stress. Finally, we demonstrate that restoration of PITPNA expression in isolated pancreatic islets from T2D human subjects rescued insulin secretory capacity and granule biogenesis and alleviated ER stress. Taken together, these results establish that diminished PITPNA function is a major cell-autonomous contributor to reduced beta-cell mass and insulin output and, ultimately, to the beta-cell failure that represents a cardinal feature of T2D pathogenesis.

## Results

### *Pitpna* is a direct target of miR-375 in the pancreatic beta-cell
The microRNA miR-375 is a potent regulator of insulin secretion that directly targets expression of several genes including *Myotrophin*, *Cadm1*, *Gephyrin (Gphn)*, and *Elavl4/HuD*[41–44]. An extended analysis using the TargetScan algorithm identified a candidate binding site for miR-375 in the 3'UTR of the gene *Pitpna*[45]. This gene encodes a

phosphatidylinositol transfer protein and is expressed in throughout the pancreatic islet in humans and mice and is not restricted to beta-cells (Fig. 1a and Supplementary Fig. 1a). Immunostaining of human pancreas for PITPNA reveals its co-localization with markers of multiple organelles including the Golgi (Giantin), the ER (Calreticulin, CALR), and intracellular vesicles (CD63) (Fig. 1a). Similarly, in mouse pancreas, Pitpna co-localized with KDEL (ER) and GM130 (Golgi) (Supplementary Fig. 1a). To determine whether *Pitpna* is a genuine miR-375 target, the full-length mouse *Pitpna* 3'UTR (2709-nt, *Pitpna* WT) was subcloned into a luciferase reporter construct and the effects of modulating miR-375 levels on reporter expression were assessed. As expected, luciferase expression was inhibited in the presence of the miR-375 mimic (375-mimic) in comparison to cells transfected with a pool of scrambled control mimics (Ctrl-mimic) (Supplementary Fig. 1b). Moreover, direct targeting of this specific site binding site was supported by our result showing that site-directed mutagenesis of the putative binding site in the 3'UTR (*Pitpna* MUT) abolished the inhibitory effect of the miR-375 mimic (Supplementary Fig. 1b). To test whether endogenous *Pitpna* expression is subject to regulation by miR-375 in vivo, murine insulinoma MIN6 cells were transfected with an inhibitory antisense RNA oligonucleotide directed against miR-375 (Antg-375) to reduce expression of this miRNA (Supplementary Fig. 1c). The Antg-375-mediated silencing of miR-375 resulted in increased *Pitpna* mRNA levels when compared to cells transfected with a control pool of scrambled antisense oligonucleotides (Antg-Ctrl) (Supplementary Fig. 1c). Immunoblot analyses confirmed that inhibition of miR-375 resulted in elevated steady-state levels of Pitpna as well as other miR-375 targets (i.e., Cadm1, Gphn) (Supplementary Fig. 1d). Conversely, transfection with the 375-mimic reduced the steady-state levels of all three of these proteins in a dose-dependent manner (Supplementary Fig. 1e). Direct binding of miR-375 with its target genes is mediated by the RNA-binding protein Argonaute2 (Ago2)[42]. Consistent with the blocking miR-375 action, conditional deletion of *Ago2* in pancreatic beta-cells (*Ins-Cre, Ago2^{flox/flox}*) de-repressed *Pitpna*, *Cadm1*, and *Gphn* expression (Supplementary Fig. 1f).

Pitpna activity represents an interesting target for miR-375 control as it is an established mediator of PtdIns-4-P synthesis within the mammalian TGN[28,29], and PtdIns-4-P is required for the recruitment of budding factors and secretory granule formation[46]. Further evidence in support of *Pitpna* expression being a physiologically relevant miR-375 target in beta-cells was provided by quantitative liquid chromatography-tandem mass spectrometry (LC/MSMS) analyses of the MIN6 murine insulinoma cell line lipidome as a function of Pitpna expression. Transfection of MIN6 cells with Antg-375 oligonucleotides to inhibit miR-375 resulted in the elevation of total bulk PtdIns in these cells as well as increased levels of multiple PtdIns molecular species (Supplementary Fig. 1g). Given the elevated insulin secretory output observed after inhibition of miR-375[41], these results suggest that *Pitpna* functional status is linked to PtdIns metabolism in the murine beta-cell.

### Decreased PITPNA expression in isolated islets of T2D human subjects
We next examined whether PITPNA functional status is an important factor in human diabetes. Datasets obtained from published transcriptomic RNA sequencing analyses performed on isolated human islet cells were interrogated for altered *PITPNA* expression[47–49]. Notably, single-cell RNA-seq analyses showed *PITPNA* expression was reduced in beta-cells from T2D donors in comparison to non-diabetic human donors with no change observed in alpha, and gamma-cell populations (Fig. 1b–d)[49]. In contrast, expression of the *PITPNB* isoform was unchanged in beta-cells of T2D human donors compared to non-diabetic donors (Supplementary Fig. 1h). Moreover, analyses of global transcriptomic RNA sequencing data from islets of human subjects stratified according to hemoglobin A1C (HbA1c) levels were also informative. HbA1c is a measure of long-term glycemia and the

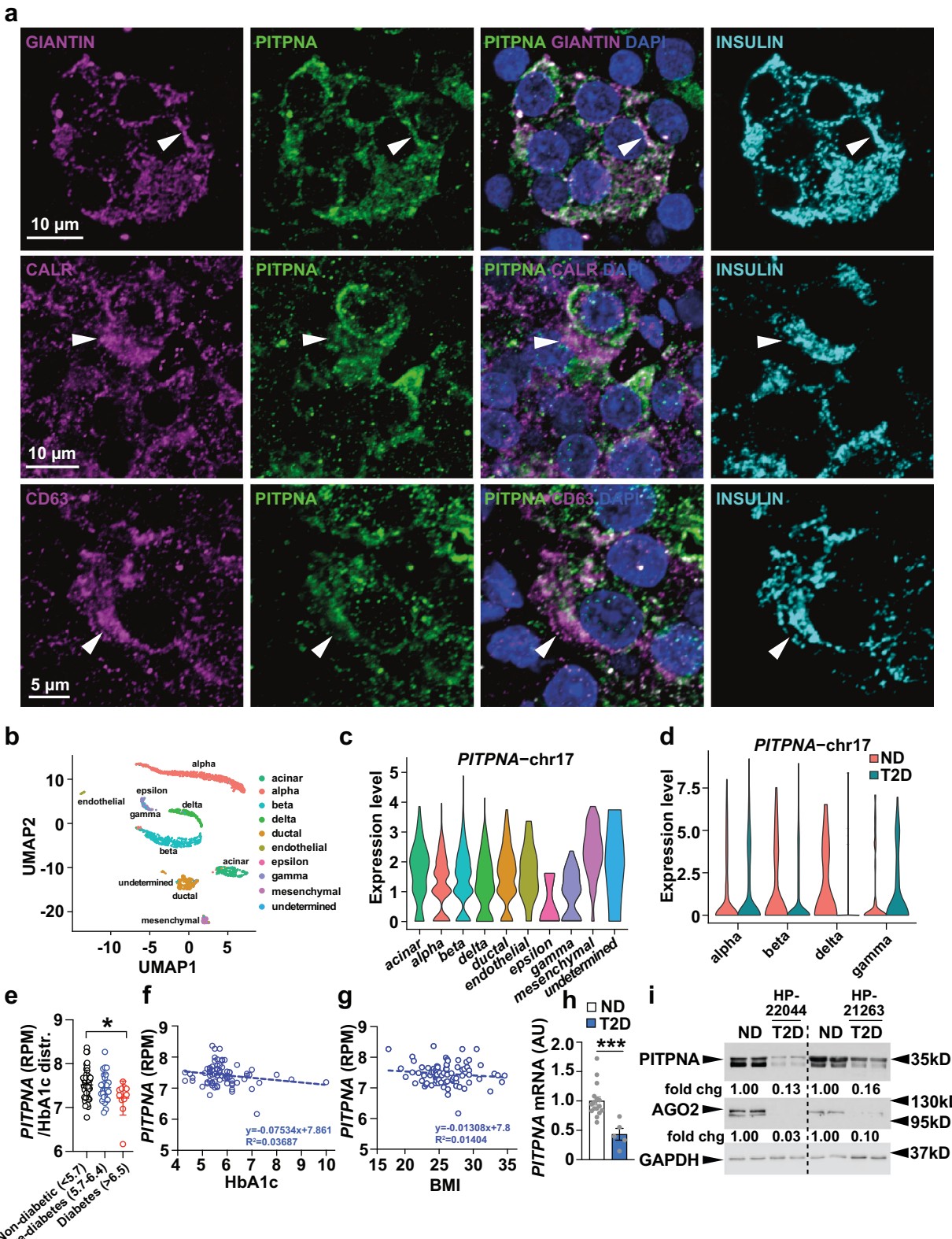

patients were classified as non-diabetic (HbA1c < 5.7%), pre-diabetic (5.7–6.4), and diabetic (>6.5)[47]. *PITPNA* gene expression (reads per million) was reduced in islets of T2D subjects (HbA1c > 6.5) relative to the expression levels recorded for islets of non-diabetic controls (HbA1c < 5.7, Fig. 1e). Indeed, *PITPNA* expression was inversely correlated with both HbA1c levels and body mass index (BMI) across all subjects (Fig. 1f, g). This inverse correlation indicates that both body

weight and glycemic status are parameters associated with changes in *PITPNA* expression. Expression analyses using quantitative real-time polymerase chain reaction (qRT-PCR) and immunoblotting further corroborated reduced PITPNA expression in isolated islets of T2D donors compared to non-diabetic (ND) control subjects (Fig. 1h, i and Supplementary Table 1). We also observed a reduction in AGO2 levels in islets of T2D donors versus ND donors (Fig. 1i) and this decrease may

**Fig. 1 | PITPNA expression is decreased in isolated islets of T2D human subjects.**
**a** Immunostaining of endogenous INSULIN (cyan), PITPNA (green), and protein markers (magenta) for the Golgi (GIANTIN), ER (CALR), and intracellular granules (CD63) in pancreatic islets isolated from a non-diabetic human subject. White arrows denote the colocalization of protein markers with PITPNA. **b** UMAP projection and graph-based clustering of scRNA-Seq analysis performed on isolated human pancreatic islet cell types. **c** Relative abundance of PITPNA in islet cell clusters from human donors. **d** Comparison of PITPNA expression in islet endocrine cell types from T2D (green) and non-diabetic donors (red). **e** Normalized PITPNA expression from bulk RNA sequencing of isolated human islets across non-diabetic ($n = 51$), pre-diabetic ($n = 27$), and diabetic human subjects ($n = 11$). Normalized expression values are shown in reads per million (RPM). $P_{Non\ vs\ Pre-diabetes} = 0.7248$, $P_{Non\ vs\ Diabetes} = 0.025$, $P_{Diabetes\ vs\ Pre-diabetes} = 0.1278$, **f** Correlation analysis between normalized islet PITPNA expression and HbA1c of human subjects ($n = 77$). The R2 value indicates the correlation coefficient. **g** Correlation analysis between normalized islet PITPNA expression and body mass index (BMI) of human subjects ($n = 89$). The R2 value indicates the correlation coefficient. **h** qRT-PCR analysis of PITPNA mRNA expression in pancreatic islets isolated from non-diabetic ($n = 15$) and T2D ($n = 5$) human donors. $P = 0.0006$. Error bar: ND = $1.00 \pm 0.02$, T2D = $0.43 \pm 0.04$. **i** Western blot analysis of PITPNA and AGO2 expression in isolated islets of non-diabetic human donors (ND) and T2D donors (T2D). Data are presented as mean values ± SEM for (**e**), (**h**). *$P < 0.05$, ***$P < 0.001$. Ordinary one-way ANOVA with Turkey's multiple comparisons test was used for (**e**). Linear regression was used for (**f**), (**g**). Two-tailed unpaired Student t-test were used for (**h**). All primary source data are reported in the Source data file.

reflect a reduced capacity for compensatory proliferation or an adaptive response by the beta-cell to increase secretory output[42,43]. Interestingly, shRNA-mediated knockdown of *PITPNA* in ND human islets also lowered AGO2 expression and indicate the reduction in these proteins are linked (Supplementary Fig. 1i). These collective data demonstrate that reduced *PITPNA* expression in pancreatic beta-cells of human subjects is associated with several hallmarks of predisposition to T2D.

## Mice lacking *Pitpna* exhibit decreased pancreatic beta-cell mass

One of the signature phenotypes of *Pitpna* whole-body knockout mice is reduced pancreatic islet numbers marked by shrunken islet morphologies and vacuolations[50]. As the majority of *Pitpna* global knockout (*Pitpna* KO) mice die within the first 48 h after birth, pancreata were isolated from these animals within the first 24 h of birth and subjected to islet morphometric analysis. In addition to quantifying the reduction in overall islet number, we observed that the number of insulin⁺ cells per area of pancreas (mm²) appeared reduced in *Pitpna* KO mice compared to littermate controls (Supplementary Fig. 2a, b). These reductions in beta-cell number were accompanied by a proportional decline in total pancreatic insulin content (Supplementary Fig. 2c). By contrast, proinsulin levels were elevated in whole pancreas lysates derived from *Pitpna* null mice relative to controls and these data indicate that loss of *Pitpna* expression compromised the relative efficiency of proinsulin processing for insulin storage (Supplementary Fig. 2c). Meanwhile, no difference was observed in either pancreatic alpha or delta cell numbers in *Pitpna* KO in comparison to littermate control mice (Supplementary Fig. 2d). Analysis of *Pitpna* null beta-cells by transmission electron microscopy (TEM) showed reduced numbers of docked granules at the plasma membrane, a reduction in the number of mature granules, and reduced overall granule size in the mutant islets relative to littermate controls (Supplementary Fig. 2e–g). Terminal nucleotidyl transferase dUTP nick end labeling (TUNEL) experiments revealed a significant increase in the number of beta-cells undergoing apoptosis in *Pitpna* whole-body knockout pancreas compared to controls (Supplementary Fig. 2h, i). The apoptotic phenotype was cell-specific as no changes were observed in TUNEL staining of the glucagon⁺ cell population (Supplementary Fig. 2j), or in Ki-67⁺ beta-cell numbers in *Pitpna* null mice (Supplementary Fig. 2k, l).

## Conditional deletion of *Pitpna* in beta-cells impairs glucose-stimulated insulin secretion (GSIS)

To more specifically assess the role of Pitpna in beta-cell physiology, two approaches were taken. First, we transfected MIN6 cells with either a scrambled siRNA control pool (si-*Ctrl*) or an siRNA pool targeting *Pitpna* (si-*Pitpna*) designed to achieve a reduction in Pitpna expression at least as great as that seen in the beta-cells of T2D islets. Transfected cells were subsequently treated with glucose in concentrations ranging from 5.5 mM to 25 mM and insulin secretion responses were measured. Indeed, insulin release in response to 10- and 25-mM glucose was markedly reduced in MIN6 cells inhibited for *Pitpna*

expression (Supplementary Fig. 3a–c). Likewise, intracellular insulin content was also significantly reduced in MIN6 cells incubated in 25 mM glucose—indicating Pitpna contributes to insulin expression, its processing and/or insulin granule biogenesis (Supplementary Fig. 3d). Conversely, transfection of MIN6 cells with an expression construct encoding the *Pitpna* cDNA increased cellular *Pitpna* expression and elevated both GSIS and insulin content in cells challenged with 25 mM glucose (Supplementary Fig. 3e–h).

Second, conditional beta-cell-specific *Pitpna* null mice were generated by crossing *Pitpna*-floxed animals with mice expressing Cre recombinase under control of the mouse *Insulin1* promoter (*Ins*-Cre, *Pitpna*^flox/flox mice)[51]. Immunoblot analyses confirmed a significant reduction in Pitpna expression in isolated islets of *Ins*-Cre, *Pitpna*^flox/flox mice by age 8 weeks (Fig. 2a). While blood glucose levels were unchanged after a 16-h fast, *Ins*-Cre, *Pitpna*^flox/flox mice exhibited elevated random-fed blood glucose in addition to reduced plasma insulin levels relative to control animals (Fig. 2b). Moreover, *Ins*-Cre, *Pitpna*^flox/flox mice exhibited reduced plasma insulin and elevated blood glucose levels in response to an intraperitoneal glucose bolus (Fig. 2c, d). These data identify impaired insulin secretion in *Pitpna*-deficient mice in comparison to control littermate mice. These results were further corroborated by a significant blunting of the ex vivo secretory response to both 16.7 mM glucose and 40 mM KCl in *Pitpna*-null islets relative to control islets (Fig. 2e). High resolution analyses of granule morphology by TEM reported increased numbers of immature and empty insulin secretory granules and reductions in the numbers of both mature secretory granules and docked granules in *Ins*-Cre, *Pitpna*^flox/flox islets (Fig. 2f–h). In addition, we observed distention of the ER and Golgi in pancreata of *Pitpna*-deficient mice (Fig. 2j). Genetic ablation of *Pitpna* in beta-cells also showed significant reductions in islet and beta cell numbers and beta-cell mass (Fig. 2k, l). These declines were in part attributed to apoptotic cell death as TUNEL staining of pancreata of 8-week-old *Ins*-Cre, *Pitpna*^flox/flox mice showed an increase in the number of TUNEL⁺ beta-cells relative to littermate control mice (Fig. 3a, b). No significant change in TUNEL staining was detected in Gcg⁺ alpha cells of *Ins*-Cre, *Pitpna*^flox/flox mice (Fig. 3c). Together these results demonstrate that Pitpna: (i) is a potent regulator of beta-cell viability, (ii) is required for insulin granule maturation and secretion in beta-cells, and (iii) beta-cell *Pitpna* deficiency is sufficient to disrupt systemic glucose homeostasis in an animal model.

## Endoplasmic reticulum (ER) stress in *Pitpna*-deficient beta-cells

Previous analysis in murine *Pitpna* null embryonic fibroblasts showed increased expression of the ER stress marker C/EBP homologous protein CHOP[50] and elevated levels of this protein are consistent with ER stress-induced apoptosis[52]. Indeed, in isolated islets of conditional *Pitpna* knockout mice, we observed elevated basal CHOP levels at steady state and upon challenge with hydrogen peroxide—an established inducer of oxidative and ER stress-mediated apoptosis (Fig. 3d)[13,53]. Furthermore, steady-state levels of the unfolded protein

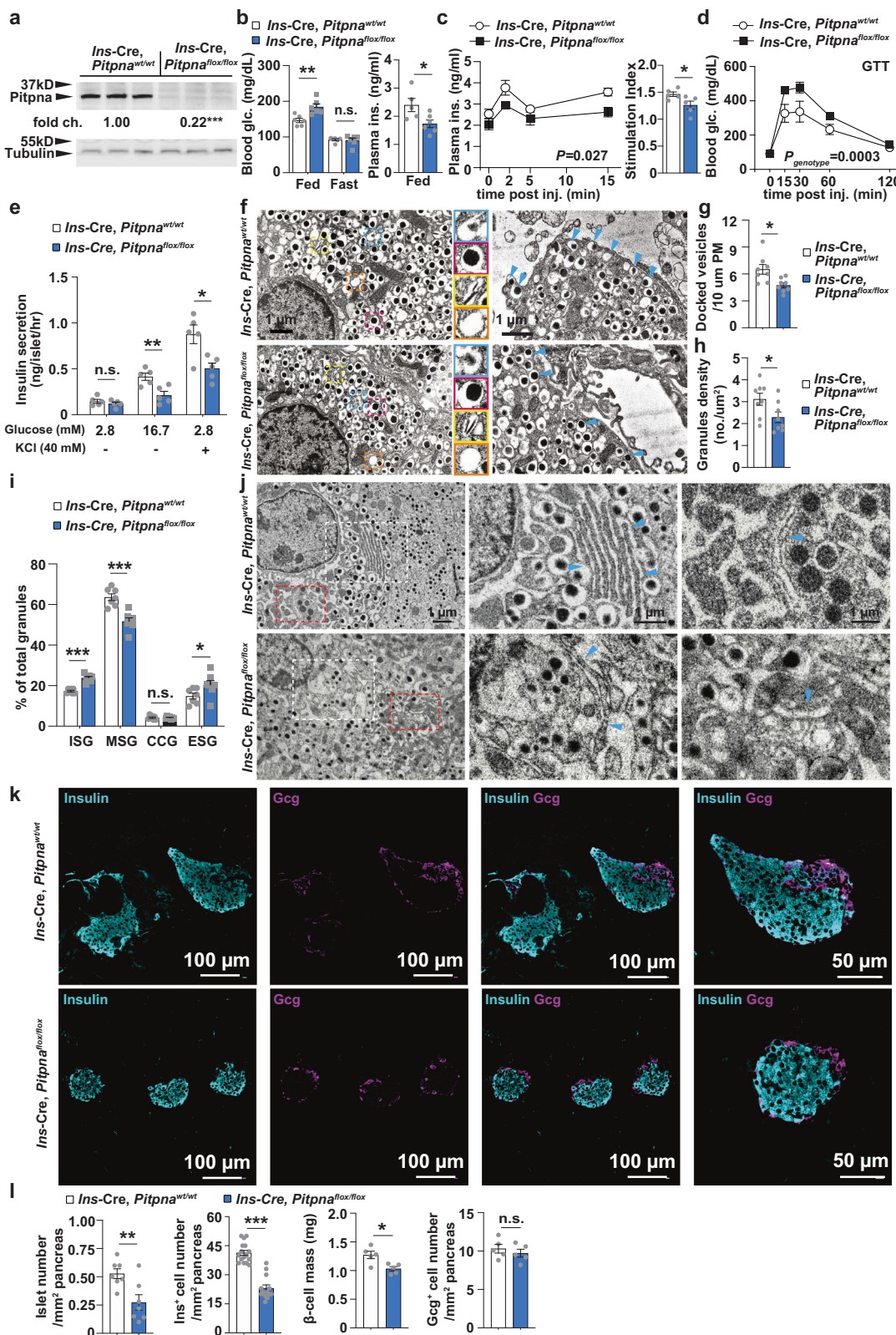

response regulator GRP78/BiP were also increased ~2-fold in *Pitpna* null islets. These observations confirm that *Pitpna*-deficient beta-cells experience elevated chronic ER stress. With regard to insulin processing, qRT-PCR analysis of the insulin processing enzymes in isolated islets of *Ins*-Cre, *Pitpna^flox/flox* mice revealed significant reductions in islet expression of *Proprotein convertase-1* (*PC1/3*), *Proprotein convertase-2* (*PC2*), and *Carboxypeptidase E* (*CPE*) (Fig. 3e). These

collective results report that Pitpna signaling is an essential component for maintaining beta-cell homeostasis. Chronic impairment of granule formation, maturation, and docking consequently triggers a cascade of ER stress and ultimately apoptosis. These defects represent the basis for the hyperglycemia observed in *Ins*-Cre, *Pitpna^flox/flox* mice and illustrate the critical roles that this lipid transfer protein executes in the beta-cell.

**Fig. 2 | Conditional deletion of Pitpna in the pancreatic beta-cell impairs glucose-stimulated insulin secretion. a** Western blot analysis of Pitpna in isolated islets from *Ins*-Cre, *Pitpna^{flox/flox}*, and littermate control *Ins*-Cre, *Pitpna^{wt/wt}* (WT) mice at age 8 weeks ($n = 3$), $P < 0.0001$. Metabolic parameters were assessed in *Ins*-Cre, *Pitpna^{flox/flox}*, and WT male mice at age 8 weeks including: **b** random-fed and over-night 16-h fasted blood glucose and plasma insulin ($n = 6$), $P = 0.0042$. **c** plasma insulin and stimulation index after glucose bolus ($n = 6$), $P_{GSIS} = 0.0272$, $P_{Stimulation\ index} = 0.0465$. **d** blood glucose measurements after glucose bolus ($n = 6$), $P_{genotype} = 0.0003$. **e** Quantification of insulin release in response to 2.8 mM and 16.7 mM glucose concentrations and KCl (40 mM) from isolated islets of 10-week-old female *Ins*-Cre, *Pitpna^{flox/flox}*, and WT mice ($n = 5$), $P_{2.8} = 0.3695$, $P_{16.7} = 0.008$, $P_{Kcl} = 0.0132$. **f** Representative transmission electron micrographs of pancreatic beta-cells from 10-week-old *Ins*-Cre, *Pitpna^{flox/flox}*, and WT female mice. **g–i** Quantification of docked vesicles (**g**, $P = 0.01$), granules density (**h**, $P = 0.0402$), granule morphology (**i**) (immature secretory granule (ISG, blue box), mature secretory granules (MSG, red box), crystal-containing granules (CCG, yellow box), and empty secretory granules (ESG, orange box) in beta-cells of 10-week-old female *Ins*-Cre, *Pitpna^{flox/flox}* ($n = 8$), and WT mice ($n = 8$) shown in panel (**f**), $P_{ISG} < 0.0001$,

$P_{MSG} < 0.0001$, $P_{CCG} = 0.746$, $P_{ESG} = 0.0274$. **j** Representative transmission electron micrographs of ER morphology and Golgi morphology of beta-cells from 10-week-old female *Ins*-Cre, *Pitpna^{flox/flox}*, and WT mice. White and red dashed boxes in the left panel identify high-magnification images of ER (center panel) and Golgi (right panel). **k** Immunostaining of insulin and glucagon (Gcg) in paraffin-embedded pancreata from 10-week-old female *Ins*-Cre, *Pitpna^{flox/flox}*, and WT mice. **l** Islet morphometric analysis including islet number per area pancreas (mm²), insulin⁺ cells per area pancreas, glucagon⁺ cells per area pancreas, and pancreatic beta-cell mass in 10-week-old female *Ins*-Cre, *Pitpna^{flox/flox}* and WT mice (WT, $n = 7, 16, 5, 5$ for islets number, insulin⁺ number, beta cell mass, Gcg⁺ number, respectively), (*Ins*-Cre, *Pitpna^{flox/flox}*, $n = 7, 13, 5, 5$ for islets number, insulin⁺ number, beta cell mass, Gcg⁺ number, respectively), $P_{insulin^+\ Nr.} < 0.0001$, $P_{islets\ Nr.} = 0.0084$, $P_{beta-cell\ mass} = 0.0128$, $P_{Gcg+\ Nr.} = 0.4698$. Data are presented as mean values ± SEM for (**b**), (**c**), (**d**), (**e**), (**g**), (**h**), (**i**), (**l**). *$P < 0.05$, ***$P < 0.001$ and n.s. denotes not significant. Two-way repeated-measure ANOVA with Post-hoc multiple comparisons test (Sidak's) was used for (**c**), (**d**). Two-tailed unpaired Student t-test were used for (**a**), (**b**), (**c**), (**e**), (**g**), (**h**), (**i**), (**l**). All primary source data are reported in the Source data file.

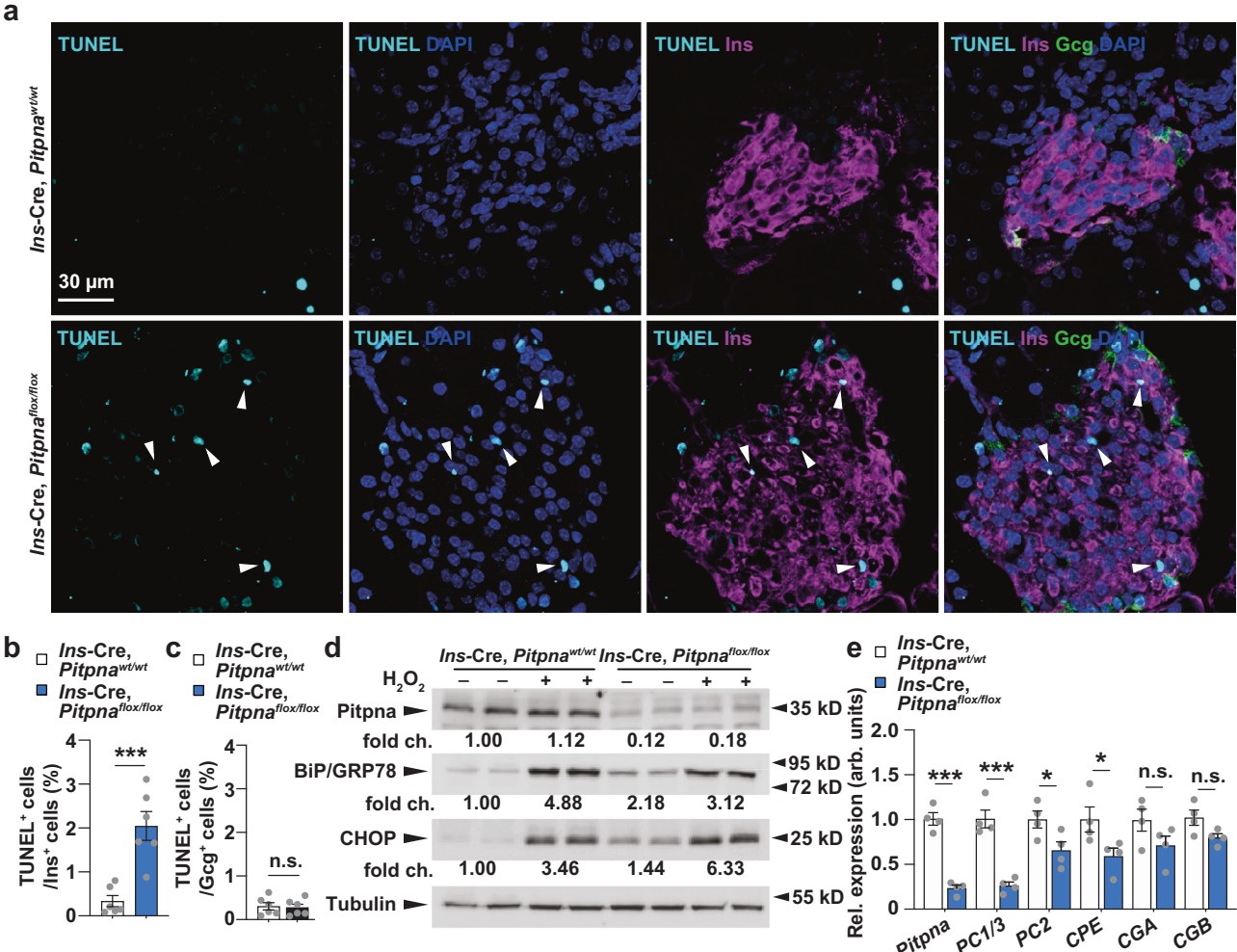

**Fig. 3 | Loss of Pitpna increases beta-cell apoptosis and expression of endoplasmic reticulum (ER) stress markers. a** Immunostaining in paraffin-embedded pancreata from 8-week-old female Ins-Cre, Pitpna^{flox/flox}, and littermate control wild-type (WT) mice was performed to assess: insulin (magenta), glucagon (Gcg, green) and apoptotic marker TUNEL (cyan). White arrowheads point to TUNEL-positive and insulin-positive cells in (**b**), TUNEL-positive beta cell number ($n = 6$), $P = 0.0007$. **c** TUNEL-positive alpha cell number ($n = 6$). **d** Western blot analysis of Pitpna, BiP/GRP78, and CHOP after treatment of hydrogen peroxide (H₂O₂) in

isolated islets of 8-week-old male Ins-Cre, Pitpna^{flox/flox}, and WT mice. **e** qRT-PCR analysis of *Pitpna*, *PC1/3*, *PC2*, *CPE*, *CGA*, and *CGB* mRNA expression in islets of male WT and Ins-Cre, Pitpna^{flox/flox} mice at age 10 weeks ($n = 5$), $P_{Pitpna} < 0.0001$, $P_{PC1/3} < 0.0001$, $P_{PC2} = 0.0441$; $P_{CPE} = 0.0488$, $P_{CGA} = 0.1313$, $P_{CGB} = 0.0585$. Data are presented as mean values ± SEM for (**b**), (**c**), (**e**). *$P < 0.05$, ***$P < 0.001$ and n.s. denotes not significant. Two-tailed unpaired Student t-test were used in for (**b**), (**c**), (**e**). All primary source data are reported in the Source data file.

## Pitpna regulates mitochondrial morphology in pancreatic beta-cells

The ER and the mitochondria engage in close physical contacts that are dynamic and are components of an inter-organelle communication system that responds to the metabolic demands of the cell[54]. In that regard, we observed that *Pitpna*-deficiency impacts mitochondrial function as reported by oxygen consumption rates (OCR) in MIN6 cells. *Pitpna* deficiencies attenuated both basal and maximal cellular respiration (Supplementary Fig. 4a, b), and suppressed glycolytic turnover and capacity as reported by the lowered extracellular acid-ification rates (ECAR) exhibited by cells silenced for *Pitpna* expression (Supplementary Fig. 4c, d). These observations are congruent with previous studies showing that brain and liver lysates from *Pitpna* whole-body knockout mice exhibited dramatically reduced total ATP and ATP/ADP ratios[50], phenotypes also consistent with reduced mito-chondrial activity[55]. Analyses of mitochondrial morphology in beta-cells of *Ins*-Cre, *Pitpna^flox/flox^* mice established that mitochondria were markedly longer in those Pitpna-deficient beta-cells, and that the

frequencies of swollen mitochondria were also significantly increased relative to littermate control animals (Supplementary Fig. 4e–h). Pre-vious studies demonstrate the guanosine triphosphatase (GTPase) Dynamin-related protein 1 (Drp1) is recruited to MERCs where its oli-gomerization enhances mitochondria constriction and fission[56,57], and that deletion of Drp1 in beta-cells results in impaired GSIS[58]. Consistent with those findings, Drp1 expression was significantly diminished in isolated islets of *Ins*-Cre, *Pitpna^flox/flox^* mice (Supplementary Fig. 4i).

## PITPNA regulates human pancreatic beta-cell function

To experimentally assess whether PITPNA is a physiologically relevant regulator of human beta-cell function, both loss and gain-of-function approaches were taken in isolated islets of non-diabetic (ND) human donors. Implementing lentiviral constructs encoding either an shRNA (which encodes an siRNA targeting the human *PITPNA* mRNA sequence) or the human *PITPNA* full-length cDNA, both knockdown (sh-*PITPNA*) and over-expression (OE-*PITPNA*) conditions were vali-dated by immunoblotting methods and qRT-PCR (Fig. 4a and

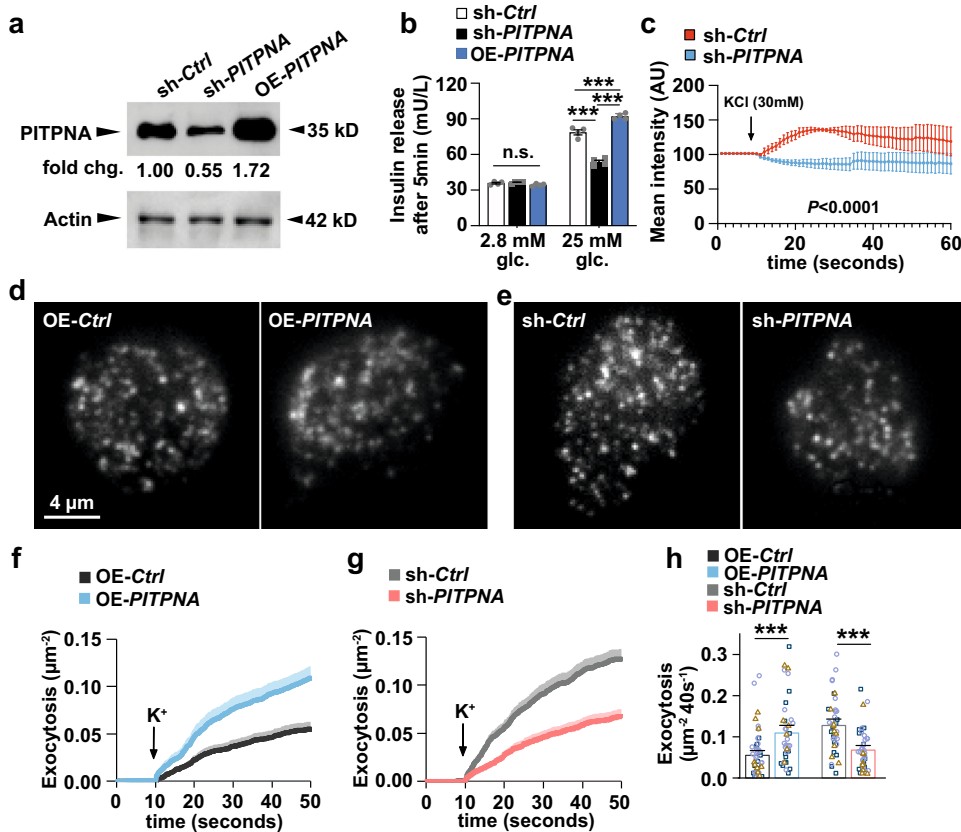

**Fig. 4 | PITPNA regulates insulin secretion in human pancreatic beta-cells.**
**a** Western blot analysis of PITPNA in isolated islets from non-diabetic human donors after treatment with lentiviruses encoding either an shRNA targeting PITPNA (sh-*PITPNA*), cDNA of human PITPNA (OE-*PITPNA*), or empty control vector (sh-*Ctrl*). **b** Quantification of insulin release in response to 2.8 mM and 25 mM glucose concentrations from isolated islets from non-diabetic human subjects after lentiviral-mediated over-expression of *PITPNA* (OE-*PITPNA*) or inhibition of *PITPNA* (sh-*PITPNA*) in comparison to treatment with control lentivirus (sh-Ctrl) (*n* = 4). Statistical Test: Ordinary one-way ANOVA with Turkey's multiple comparisons test. Values: For 2.8 mM glucose, Summary: F = 3.937, *P* = 0.0591; Multiple comparison: sh-Ctrl vs sh-*PITPNA*: *P* = 0.7839, sh-Ctrl vs OE-*PITPNA*: *P* = 0.1625, sh-*PITPNA* vs OE-*PITPNA*: *P* = 0.0581. For 25 mM glucose, Summary: F = 137, *P* < 0.0001; Multiple comparisons: sh-*Ctrl* vs sh-*PITPNA*: *P* < 0.0001, sh-*Ctrl* vs OE-*PITPNA*: *P* = 0.0008, sh-*PITPNA* vs OE-*PITPNA*: *P* < 0.0001. **c** Quantification of intracellular Ca²⁺ concentra-tion in isolated human islets after lentiviral-mediated inhibition of PITPNA (sh-*PITPNA*, *n* = 7) in comparison to control lentivirus (sh-*Ctrl*, *n* = 5). **d, e** Representative

TIRF images of human beta-cells expressing the granule marker NPY-tdmOrange2 after treatment with lentiviruses encoding GFP control (OE-*Ctrl*), cDNA of human *PITPNA* (OE-*PITPNA*) in panel (**d**), as well as an shRNA pool targeting *PITPNA* (sh-*PITPNA*) or shRNA control (sh-*Ctrl*) in panel (**e**); Scale bar, 4 μm. **f, g** Cumulative time course of high K⁺-evoked exocytosis events normalized to cell area, for conditions as in (**d**), (**e**). Bars at individual time points indicate SEM., K⁺ was elevated to 75 mM during *t* = 10–50 s. **h** Total exocytosis measured during TIRF analysis of human beta-cells expressing the granule marker NPY-tdmOrange2 after lentivirus treat-ments represented in panels (**d**) and (**e**). Statistical Test: Two-way ANOVA with Tukey's multiple comparisons test. Values: Summary: F = 11.67, *P* < 0.0001; Multiple comparison: sh-Ctrl vs sh-PITPNA: *P* = 0.0003, Ctrl vs OE-PITPNA: *P* = 0.001. The data set was generated from 3 unique human donors; dots and their color/symbol indicate individual cells and donors, respectively. Data presented as mean ± SEM, unless otherwise indicated. ***P* < 0.001, and n.s. denotes not significant. All primary source data are reported in the Source data file.

Supplementary Fig. 5a). GSIS was subsequently assessed in isolated human islets of ND donors after challenge with either sh-*PITPNA* or OE-*PITPNA* lentiviral vectors or control lentiviruses encoding either a non-targeting shRNA vector (sh-*Ctrl*) or green fluorescent protein (GFP) (OE-*Ctrl*). No alterations were observed in PITPNA expression in isolated human islets after treatment with control lentiviruses (Supplementary Fig. 5b). Meanwhile consistent with results from isolated islets of *Ins*-Cre, *Pitpna*^*flox/flox* mice, *PITPNA* knockdown inhibited insulin secretion upon stimulation with 25 mM glucose, while GSIS was significantly elevated after *PITPNA* over-expression relative to mock-treated islets (Fig. 4b). In addition, intracellular [$Ca^{2+}$] was diminished in response to a 30 mM KCl stimulus in islets of ND donors where *PITPNA* expression was inhibited. These data support a model where the primary PITPNA execution point lies downstream of $K^+$ channel closure—i.e., at the level of granule trafficking, docking, and/or exocytosis (Fig. 4c).

To interrogate how PITPNA affects stimulus-secretion coupling, total internal reflection (TIRF) microscopy was used to monitor exocytosis and docking of insulin granules in dispersed human islet cells. After plating, islet cells were treated with either sh-*PITPNA* or OE-*PITPNA* lentiviral vectors or their respective control lentivirus (sh-*Ctrl* or an empty vector control, OE-*Ctrl*) (Fig. 4d, e). In addition, a genetically encoded NPY-tdmOrange2 marker was used to label granules (Fig. 4d, e). Exocytosis was evoked by depolarization with elevated $K^+$ (in the presence of diazoxide to prevent spontaneous depolarization) and it followed a biphasic time course (Fig. 4f, g). Exocytosis was increased in the face of PITPNA overexpression (by 98% vs OE-*Ctrl* cells; $P = 0.0004$ non-paired t-test; 50 OE-*Ctrl* cells and 40 OE-*PITPNA* cells; 3 donors each; Fig. 4f, h), and decreased by *PITPNA* silencing (by 47% vs sh-*Ctrl*; $P = 1E-05$ non-paired t-test; 40 sh-*Ctrl* cells and 43 sh-*PITPNA* cells; 3 donors each; Fig. 4g, h). These data indicate a positive correlation between PITPNA expression and exocytosis in human beta cells and, from these observations, we conclude that PITPNA-dependent changes in exocytosis reflect changes in the secretory machinery of individual insulin granules. Electron microscopy imaging revealed that insulin granule core density and numbers of docked vesicles were significantly reduced after *PITPNA* knockdown in isolated human ND islets relative to control lentivirus-treated islets (Fig. 5a–c). Moreover, shRNA-mediated silencing of *PITPNA* impaired the formation of mature secretory granules (MSG) with a reciprocal increase in the numbers of immature secretory granules (ISG) (Fig. 5d). Conversely, *PITPNA* over-expression in isolated ND islets increased MSG numbers with associated reductions in ISG numbers (Fig. 5d). These results demonstrate that PITPNA is a potent regulator of granule maturation and docking in human beta-cells.

The collective insulin granule data collected in both human and mouse loss-of-function studies suggested PITPNA insufficiencies in human beta-cells ultimately disrupt proinsulin packaging into insulin granules. Indeed, in a manner consistent with the results from *Ins*-Cre, *Pitpna*^*flox/flox* mice, proinsulin levels were elevated upon *PITPNA* silencing (sh-*PITPNA*) in isolated human ND islets (Fig. 5e, f). Reciprocally, islet insulin levels were reduced relative to control lentivirus-treated human ND islets (sh-*Ctrl*)—further reporting proinsulin processing is impaired upon loss of PITPNA activity. By contrast, increasing PITPNA expression in islets (OE-*PITPNA*) elevated both proinsulin and insulin levels compared to control-treated human ND islets. These results demonstrate that the enhanced GSIS supported by increased PITPNA activity is supported by increased granule maturation and proinsulin processing. Previous studies highlighted an association of proinsulin accumulation with perturbed expression of UPR/ER stress proteins[59]. Our findings with *Ins*-Cre, *Pitpna*^*flox/flox* mice (Fig. 3d) prompted examination of whether proinsulin accumulation in isolated ND human islets induces ER stress. Immunoblot analyses confirmed *PITPNA* silencing in ND human islets (sh-*PITPNA*) resulted in increased expression of CHOP as well as other components of the ER stress

pathway—including inositol-requiring enzyme 1 alpha (IRE1a) and protein disulfide isomerase-a1 (PDI) (Fig. 5g). By contrast, protein disulfide oxidase ER-Oxidoreducin 1 alpha (ERO1) steady-state levels were decreased when *PITPNA* was silenced in ND human islets. These collective data report that expression levels of multiple components of the ER stress pathway are perturbed under conditions of PITPNA insufficiency (Fig. 5g). Moreover, these collective data demonstrate that loss of *PITPNA* results in similar derangements in both human and mouse beta-cell systems. These include impaired granule biogenesis and maturation that is accompanied by increased islet expression of multiple ER stress markers such as CHOP—a member of the C/EBP family of transcription factors linked to programmed cell death[60–62]. The translation of phenotypes associated with PITPNA deficiencies from murine to human beta-cells extended to mitochondrial dysmorphologies. *PITPNA* silencing in ND human beta-cells resulted in lengthening of mitochondrial ribbons while *PITPNA* over-expression markedly shifted the morphological distribution to shorter mitochondria (Supplementary Fig. 5c, d). *PITPNA* silencing in human beta-cells also reduced the number of morphologically 'orthodox' mitochondria with proportional increases in the frequencies of 'swollen' mitochondria (Supplementary Fig. 5e). Furthermore, as observed in *Ins*-Cre, *Pitpna*^*flox/flox* mice, inhibition of *PITPNA* in ND human islets (sh-*PITPNA*) also incurred distention of the ER in comparison to control-treated islets (sh-*Ctrl*) while over-expression of *PITPNA* (OE-PITPNA) did not alter ER morphology (Supplementary Fig. 5f, g). These results demonstrate that PITPNA in human beta-cells potently regulates insulin exocytosis, intracellular $Ca^{2+}$ concentrations, granule maturation and docking, and mitochondrial and ER morphology.

## PITPNA mediates PtdIns-4-P synthesis in human pancreatic islets

The available data indicate PITPNA stimulates PtdIns 4-OH kinases by using its lipid-exchange activity to render PtdIns a better substrate for the enzyme and it is in this way that PITPNA promotes PtdIns-4-P synthesis[28,63,64]. To test whether reduction of *PITPNA* expression in human islets attenuates formation of PtdIns-4-P, we again performed shRNA-mediated silencing of *PITPNA* in isolated islets of non-diabetic donors. After 48 h of incubation, PtdIns-4-P status was assessed using immunohistochemical methods (Fig. 6a). The PtdIns-4-P signal in insulin⁺ beta-cells was significantly diminished—indicating lentiviral-mediated knockdown of *PITPNA* significantly reduced cellular levels of this phosphoinositide (Fig. 6b). These results were supported by an independent assay monitoring GOLPH3 localization as this protein is recruited to TGN membranes by virtue of its ability to bind PtdIns-4-P[28,65]. *PITPNA* silencing in isolated human islets evoked release of GOLPH3 from TGN membranes as evidenced by the significant reductions in GOLPH3 co-localization with the TGN marker GOLGIN97 (Fig. 6c, d).

To further test whether reduced *PITPNA* expression in human islets attenuates PtdIns-4-P synthesis, *PITPNA* was either silenced or over-expressed in isolated islets of ND human donors. Mass spectrometry-based quantitative lipidomics were then performed to measure bulk cellular PtdIns and PtdIns-P levels. Although mass spectrometry cannot distinguish regio-isomers; PtdIns-4-P is the most abundant isomer in mammalian cells and PtdIns-4-P is estimated to constitute >90% of total cellular PtdIns-P[66,67]. After normalization of PtdIns-P levels to total cellular PtdIns, the data demonstrate that *PITPNA* over-expression (*PITPNA-OE*) in isolated islets of ND human donors increased PtdIns-P levels compared to empty vector control transfections (OE-*Ctrl*) (Fig. 6e and Supplementary Fig. 5f). Conversely, *PITPNA* silencing in isolated human islets decreased cellular PtdIns-P levels relative to control islets (Fig. 6f and Supplementary Fig. 5g). These observations demonstrate that modulation of *PITPNA* in isolated human islets impacts PtdIns-4-P homeostasis, and are consistent with studies in mammalian neural stem cells[28].

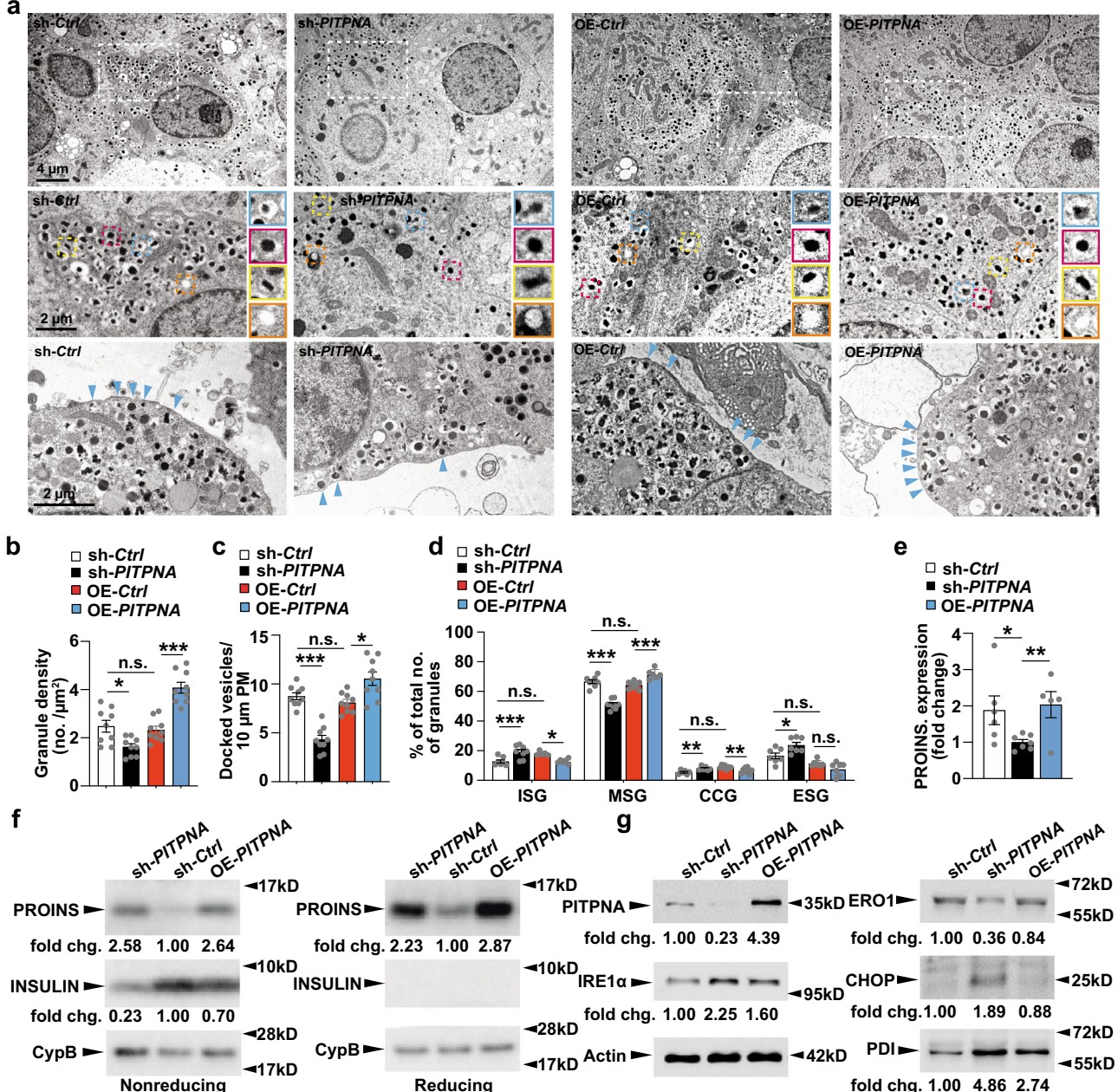

**Fig. 5 | PITPNA regulates insulin granule maturation and proinsulin processing in human pancreatic beta-cells. a** Representative transmission electron micrographs of pancreatic beta-cells from non-diabetic human donors after lentiviral-mediated over-expression of *PITPNA* (OE-*PITPNA*) or inhibition of *PITPNA* (sh-*PITPNA*) in comparison to respective control lentiviruses (OE-*Ctrl* or sh-*Ctrl*). The second raw image represents a zoom-in view of the white box from the first raw image. Granule profile: immature secretory granules (blue box), mature secretory granules (red box), crystal-containing granules (yellow box), and empty secretory granules (orange box). **b** Quantification of granule density in beta-cells of lentiviral-treated human islets shown in panel (**a**) ($n = 9$). $P_{sh\text{-}Ctrl\ vs\ sh\text{-}PITPNA} = 0.0199$, $P_{OE\text{-}Ctrl\ vs\ OE\text{-}PITPNA} < 0.0001$. **c** Quantification of docked vesicles in beta-cells of lentiviral-treated human islets shown in panel (**a**) ($n = 9$). $P_{sh\text{-}Ctrl\ vs\ sh\text{-}PITPNA} = 0.0004$, $P_{OE\text{-}Ctrl\ vs\ OE\text{-}PITPNA} = 0.00421$. **d** Quantification of immature secretory granules (ISG, $n = 7, 8, 5, 6$ for sh-*Ctrl*, sh-*PITPNA*, OE-*Ctrl*, OE-*PITPNA*, respectively), mature secretory granules (MSG, $n = 8, 8, 8, 6$ for sh-*Ctrl*, sh-*PITPNA*, OE-*Ctrl*, OE-*PITPNA*, respectively), crystal-containing granules (CCG, $n = 6, 8, 8, 8$ for sh-*Ctrl*, sh-*PITPNA*, OE-*Ctrl*, OE-*PITPNA*, respectively), and empty secretory granules (ESG, $n = 7, 7, 7, 7$ for sh-*Ctrl*,

sh-*PITPNA*, OE-*Ctrl*, OE-*PITPNA*, respectively) in beta-cells of isolated human islets after lentiviral treatments shown in panel (**a**). ($P_{sh\text{-}Ctrl\ vs\ sh\text{-}PITPNA} < 0.0001$, $P_{OE\text{-}Ctrl\ vs\ OE\text{-}PITPNA} = 0.0128$ for ISG), ($P_{sh\text{-}Ctrl\ vs\ sh\text{-}PITPNA} < 0.0001$, $P_{OE\text{-}Ctrl\ vs\ OE\text{-}PITPNA} = 0.0008$ for MSG), ($P_{sh\text{-}Ctrl\ vs\ sh\text{-}PITPNA} = 0.0068$, $P_{OE\text{-}Ctrl\ vs\ OE\text{-}PITPNA} = 0.6874$ for CCG), ($P_{sh\text{-}Ctrl\ vs\ sh\text{-}PITPNA} = 0.0207$, $P_{OE\text{-}Ctrl\ vs\ OE\text{-}PITPNA} = 0.4262$ for ESG). **e** Proinsulin expression in isolated human islets after lentiviral-mediated over-expression of *PITPNA* (OE-*PITPNA*, $n = 5$ biologically independent samples), knockdown of *PITPNA* (sh-*PITPNA*, $n = 7$ biologically independent samples) or control lentivirus (sh-*Ctrl*, $n = 7$ biologically independent samples). **f** Quantification of proinsulin in isolated human islets after densitometric analysis of western blots shown in panel (**f**). **g** Western blot analysis of PITPNA, and ER stress/unfolded protein response (UPR) proteins IRE1α, ERO1, PDI, and CHOP in human islets after lentiviral-mediated over-expression of PITPNA (OE-*PITPNA*), knockdown of PITPNA (sh-*PITPNA*) or control lentivirus (sh-*Ctrl*). Data are presented as mean values ± SEM for (**b**), (**c**), (**d**), (**e**). *$P < 0.05$, **$P < 0.01$, ***$P < 0.001$. Ordinary one-way ANOVA with Turkey's multiple comparisons test was used for (**b**), (**c**), (**d**), (**e**). All primary source data are reported in the Source data file.

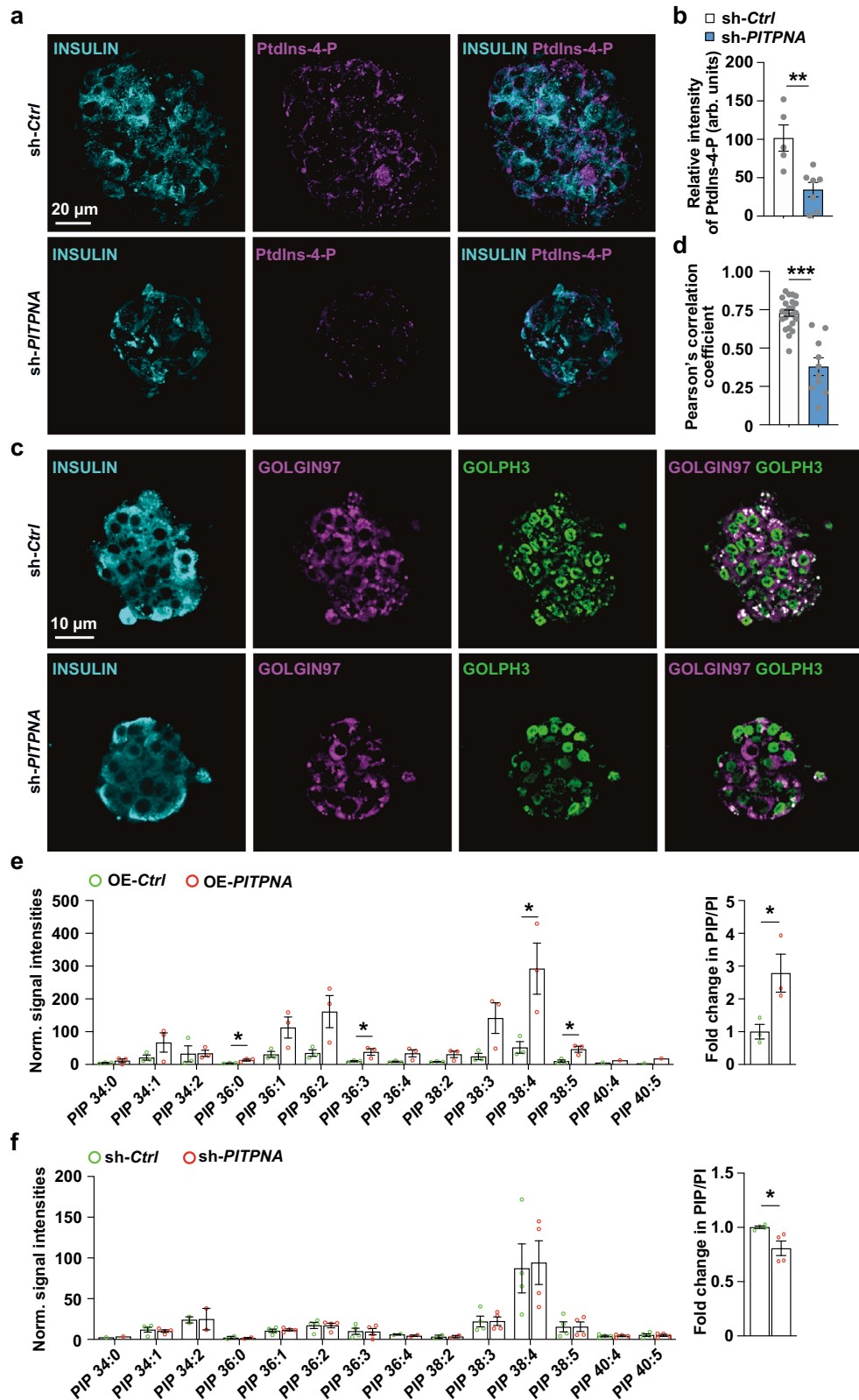

### Restoration of PITPNA rescues beta-cell function in T2D islets

The weight of the collective data gleaned from both murine animal models and human islets, including the demonstration that *PITPNA* expression was diminished in pancreatic beta-cells of T2D human subjects, implicates PITPNA as a major factor in beta-cell failure during T2D. These aggregate results raised the provocative question of whether recovery of PITPNA expression in T2D islets restores function to the diseased tissue. Indeed, lentiviral-mediated induction of *PITPNA* in isolated islets from T2D human donors significantly elevated PITPNA expression levels in islets of the T2D donor (T2D-*PITPNA* OE), and these levels were comparable to the endogenous expression levels recorded for islets of the ND human donor (ND) (Fig. 7a). Strikingly, the rescue of *PITPNA* expression in T2D islets significantly improved GSIS in response to 15 mM glucose compared to respective control-treated

**Fig. 6 | Inhibition of PITPNA in isolated human islets disrupts subcellular localization of PtdIns-4-P to the TGN. a, b** Immunostaining for INSULIN and PtdIns-4-P, and quantification of the intensity of PtdIns-4-P in isolated human islets after lentiviral-mediated inhibition of PITPNA (sh-*PITPNA*, $n = 7$) or treatment with control lentivirus (sh-*Ctrl*, $n = 5$). $P = 0.0041$. Scale bar = 20 μm. **c, d** Quantification of GOLGIN97 and GOLPH3 colocalization after immunostaining of isolated human islets after lentiviral-mediated inhibition of PITPNA (sh-*PITPNA*) ($n = 10$) or treatment with control lentivirus (sh-*Ctrl*) ($n = 23$). $P < 0.0001$. Scale bar = 10 μm. **e** Quantification of phosphatidylinositol-phosphate (PtdIns or PIP) species in isolated human islets after lentiviral-mediated over-expression of PITPNA (OE-*PITPNA*) or treatment with a control lentivirus (OE-*Ctrl*) ($n = 3$). Results normalized to total cellular PtdIns (PI). $P_{PIP34:0} = 0.3085$, $P_{PIP34:1} = 0.2069$, $P_{PIP34:2} = 0.9604$, $P_{PIP36:0} = 0.0432$, $P_{PIP36:1} = 0.0703$, $P_{PIP36:2} = 0.0646$, $P_{PIP36:3} = 0.05$, $P_{PIP36:4} = 0.0788$, $P_{PIP38:2} = 0.0815$, $P_{PIP38:3} = 0.069$, $P_{PIP38:4} = 0.0394$, $P_{PIP38:5} = 0.3604$, $P_{fold\ change} = 0.045$. **f** Quantification of phosphatidylinositol-phosphate (PtdIns or PIP) species in isolated human islets after lentiviral-mediated inhibition of PITPNA (sh-*PITPNA*) or treatment with a lentivirus expressing an shRNA control (sh-*Ctrl*) ($n = 4$). Results normalized to total cellular PtdIns (PI). $P_{PIP34:1} = 0.6432$, $P_{PIP34:2} = 0.958$, $P_{PIP36:0} = 0.793$, $P_{PIP36:1} = 0.6891$, $P_{PIP36:2} = 0.9853$, $P_{PIP36:3} = 0.9284$, $P_{PIP36:4} = 0.3006$, $P_{PIP38:2} = 0.9485$, $P_{PIP38:3} = 0.9601$, $P_{PIP38:4} = 0.8663$, $P_{PIP38:5} = 0.9815$, $P_{PIP40:4} = 0.8955$, $P_{PIP40:5} = 0.8219$, $P_{fold\ change} = 0.0284$. Data are presented as mean values ± SEM for (**b**), (**d**), (**e**), (**f**). *$P < 0.05$, **$P < 0.01$, ***$P < 0.001$, and n.s. denotes not significant. Two-tailed unpaired Student t-test were used in for (**b**), (**d**), (**e**), (**f**). All primary source data are reported in the Source data file.

T2D islets (Fig. 7b and Supplementary Fig. 6a). Moreover, recovery of PITPNA expression in T2D islets rescued PtdIns-4-P synthesis as evidenced by PITPNA inducing redistribution of GOLPH3 from a dispersed cytoplasmic localization to TGN membranes marked by GOLGIN97 (Fig. 7c, d).

Restoration of PITPNA expression to T2D islets exhibited other profound effects. Electron microscopy analyses indicated insulin granule number, docking, and maturation were rescued upon induction of *PITPNA* expression in T2D islets (Fig. 7e). Notably, insulin granule number per μm² was significantly lower in T2D islets compared to granule density in non-diabetic islets. Recovery of *PITPNA* expression in T2D beta-cells markedly rescued the reduction in granule number (Fig. 7f), fully rescued the granule docking defects in T2D beta-cells (Fig. 7g), and effected a partial rescue of mature granule numbers (Fig. 7h). In addition, restoration of PITPNA expression in islets of six individual T2D human donors (T2D-OE) resulted in the down-regulation of steady-state levels of CHOP, PDI, and BiP/GRP78 (Supplementary Fig. 6b) as well as restoration of proinsulin expression (Supplementary Fig. 6c). Taken together, these results demonstrate that restoration of *PITPNA* expression in T2D beta-cells substantially reverses the GSIS defects, the impaired insulin granule biogenesis and maturation, and the chronic ER stress associated with human T2D.

## Discussion

Critical to the development of therapeutics for diabetes are strategies for promoting insulin release while preserving pancreatic beta-cell mass. Recent studies focus on defects in insulin processing and granule maturation as causes for reduced insulin secretion that are linked to all major forms of diabetes[4,7]—a focus that rests on demonstrations that: (1) glucose-dependent granule docking is a limiting factor for insulin secretion and (2) reduced granule docking characterizes beta-cell dysfunction during human T2D[68]. In this study, we demonstrate that reduced PITPNA-dependent PtdIns-4-P signaling in the beta cell TGN results in beta-cell failure. We show that PITPNA deficiencies impair insulin granule maturation and exocytosis, and that these trafficking defects induce proinsulin accumulation, promote chronic ER stress, and derange mitochondrial dynamics and performance. The data outline a high degree of functional dependence between the TGN, ER, and mitochondria, and identify PITPNA as a central regulator of this intra-organelle crosstalk. Finally, we report the remarkable demonstration that restoring PITPNA expression to T2D human islets is sufficient to reverse beta-cell failure by rescuing GSIS, insulin granule maturation, proinsulin processing, and by alleviating the chronic ER stress that accompanies these defects in T2D beta-cells. These results: (i) highlight PITPNA-dependent PtdIns-4-P synthesis on TGN membranes as critical for sustaining insulin granule biogenesis and maturation, (ii) indicate compromise of this activity is a powerful marker of beta-cell failure during T2D, and (iii) identify new prospects for T2D therapy.

All available in vivo evidence, collected from single-cell yeast to mammalian models, indicates that soluble PITPs potentiate constitutive membrane trafficking from late compartments of the secretory pathway—specifically TGN/endosomes. Analyses of headgroup-specific PITP mutants and localization of PtdIns-4-P biosensors indicate the biochemical basis for PITP function is to stimulate PtdIns-4-P synthesis on TGN/endosomal membranes with the result that PtdIns-4-P binding proteins (i.e., effectors of PtdIns-4-P signaling) are recruited to these compartments[27–29,36–40,69,70]. It is in this fashion that PtdIns-4-P is proposed to act as a transient tag to convey spatial information that helps organize membrane trafficking[18,71]. The current demonstration that PITPNA is required for insulin granule formation, maturation, and exocytosis now extends this concept to regulated membrane trafficking pathways in human pancreatic beta-cells. This conclusion is further supported by: (i) the demonstration that modulation of *PITPNA* in human beta-cells regulates PtdIns-P, (ii) the Sac2 phosphatase is a PtdIns-4-P binding protein that localizes to the insulin granule surface where it mediates granule docking to the plasma membrane and exocytosis[72], and (iii) measurements reporting that PtdIns (the direct metabolic precursor of PtdIns-4-P) constitutes ~21% of insulin granule lipid in the INS-1 832/13 beta-cell line[73]. While the precise role(s) of PtdIns-4-P in granule docking and exocytosis remains to be fully clarified, the demonstration that dephosphorylation of PtdIns-4-P by the phosphatase Sac2 disrupts insulin granule docking and GSIS, and that Sac2 expression is decreased in T2D islets alludes to its functional significance[72,74]. We suggest that Sac2-mediated dephosphorylation of PtdIns-4-P 'signals' the end of the insulin granule biogenesis/maturation phase, and 'identifies' the mature granule as competent for mobilization to the plasma membrane for docking and exocytosis.

A striking consequence associated with PITPNA inhibition in human beta-cells is the potent increase in proinsulin levels. Initial accumulation of proinsulin correlates with a stressed ER in islets of *Lepr^db/db* mice as blood glucose levels rise (~237 mg/dL), and is maintained until proinsulin levels dramatically fall upon onset of severe hyperglycemia (~523 mg/dL)[75]. Our demonstration that PITPNA levels are significantly reduced in T2D islets compared to expression in islets of non-diabetic controls, and that restoring PITPNA expression to the beta-cell helps to recover proinsulin expression, agree with those previous findings. The accumulation of proinsulin detected after acute inhibition of *PITPNA* in human islets may reflect the impaired granule formation and/or maturation at an early stage of dysfunction, that persists until chronic insulin demand and ER stress cause the beta-cell to cease proinsulin production leading to hyperglycemia. Moreover, the accumulation of proinsulin after acute inhibition of *PITPNA* shows downstream defects in granule maturation and docking and GSIS are ultimately linked to induce ER stress[76]. Activation of the ER stress pathway might be directly related to adverse changes in mitochondrial or ER dynamics[11,52], or in activation of an inter-organellar response that negatively feeds back on proinsulin export from the ER[77].

The perturbations in mitochondrial performance and health of the endoplasmic reticulum in PITPNA-deficient beta-cells are notable. PtdIns-4-P is present on the surface of TGN-derived vesicles recruited

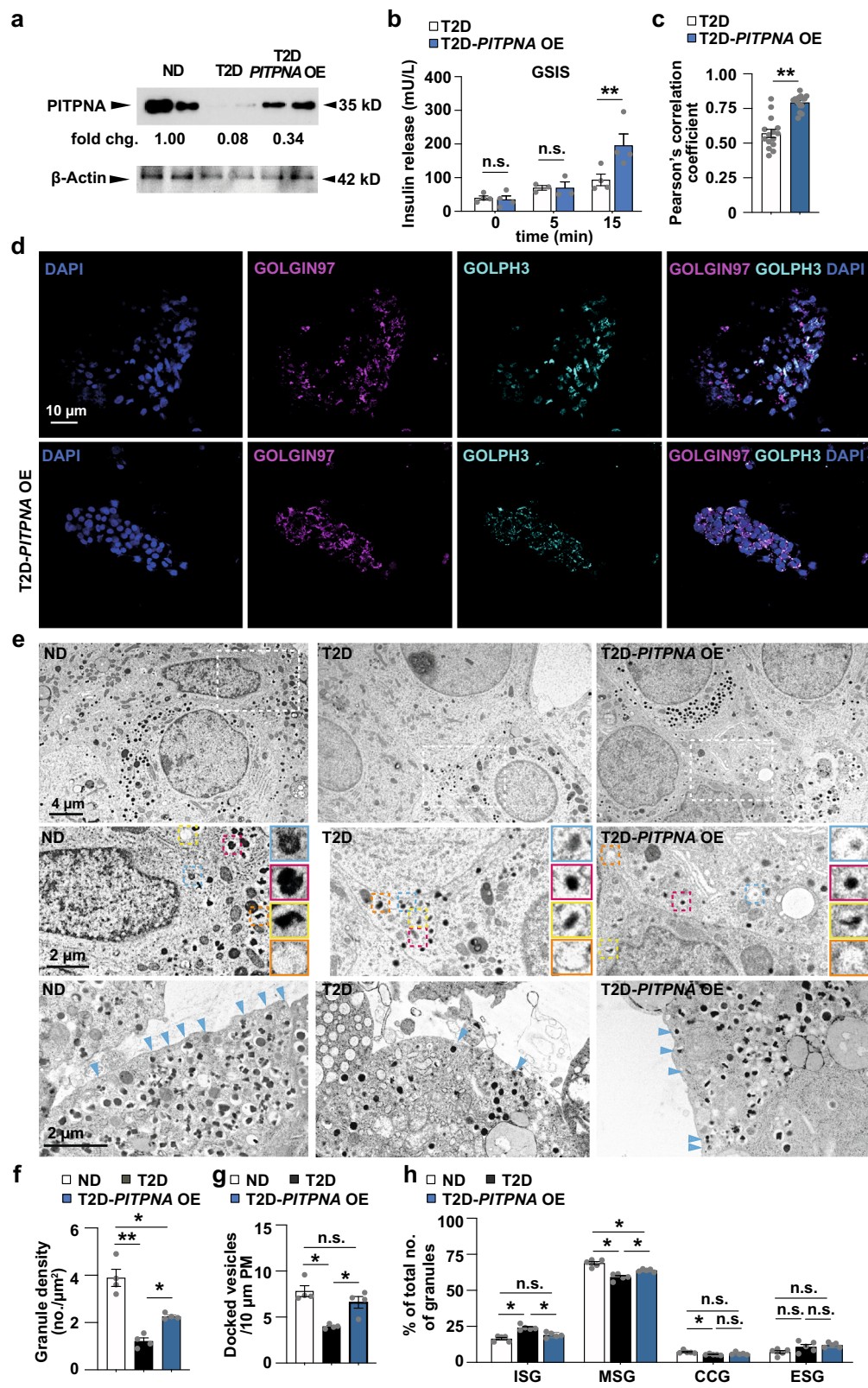

to MERCs, and this PtdIns-4-P pool is reported to aid in potentiation of mitochondrial fission and ER dynamics[54,78–81]. As PITPNA promotes PtdIns-4-P synthesis in the mammalian TGN by facilitating presentation of PtdIns to PtdIns 4-OH kinases[28], we suggest that the PITPNA-regulated PtdIns-4-P pool in beta-cells the coordinates actions of the TGN in ER/mitochondrial dynamics in addition to facilitating insulin granule biogenesis. It is presently thought that mitochondrial fission is

essential for sustaining a healthy pool of mitochondria by allowing for the clearance of damaged mitochondria through mitophagy and de novo biogenesis[79,82]. Moreover, mitophagy protects human pancreatic beta-cells from inflammatory damage during diabetes[83]—indicating the removal of dysfunctional mitochondria is essential for preventing inflammatory stress and cell death. Our results showing mitochondrial lengthening as a consequence of functional ablation of *PITPNA* in both

**Fig. 7 | Restoration of PITPNA in isolated islets of T2D human subjects rescues pancreatic beta-cell function. a** Western blot analysis of PITPNA in human non-diabetic (ND), and T2D islets after either lentiviral-mediated over-expression of PITPNA (T2D-*PITPNA* OE) or treatment with a control lentivirus (T2D) (*n* = 2). **b** Quantification of insulin release from isolated islets from T2D donors after either lentiviral-mediated over-expression of PITPNA (T2D-*PITPNA* OE) or treatment with a control lentivirus (T2D) (*n* = 4). $P_0 = 0.0284$, $P_5 = 0.7128$, $P_{15} = 0.0363$. **c**, **d** Quantification of GOLGIN97 and GOLPH3 colocalization after immunostaining of isolated human islets from T2D donors after either lentiviral-mediated over-expression of PITPNA (T2D-*PITPNA* OE) or treatment with a control lentivirus (T2D) (*n* = 15). *P* < 0.0001. **e** Representative transmission electron micrographs of pancreatic beta-cells of non-diabetic (ND) or T2D human donors after treatment with a control lentivirus (T2D) or lentiviral-mediated over-expression of PITPNA (T2D-*PITPNA* OE). The second raw image represents a zoom-in view of the white box from the first raw image. Immature secretory granules (blue box), mature secretory granules (red box), crystal-containing granules (yellow box), and empty secretory granules (orange box). Blue arrows identify docked granules. **f** Quantification of granule density, **g** docked vesicles, and **h** granule profile: immature secretory granule (ISG), mature secretory granules (MSG), crystal-containing granules (CCG), and empty secretory granules (ESG) in beta-cells of non-diabetic (ND) or T2D human donors after treatment with a control lentivirus (T2D) or lentiviral-mediated over-expression of PITPNA (T2D-*PITPNA* OE) (*n* = 4). Data are presented as mean values ± SEM for (**b**), (**c**), (**f**), (**g**), (**h**). *P < 0.05, **P < 0.01, ***P < 0.001. Ordinary one-way ANOVA with Turkey's multiple comparisons test was used for (**f**), (**g**), (**h**). Two-tailed unpaired Student t-test were used in for (**b**), (**c**). All primary source data are reported in the Source data file.

murine and human beta-cells suggests diminished PITPNA-dependent PtdIns-4-P synthesis impacts mitochondrial dynamics in the beta-cell. That *PITPNA* insufficiencies in human beta-cells induce accumulation of swollen mitochondria further emphasizes this point.

Our demonstration that *Pitpna* is a direct target of miR-375 shows that the complex relationship between the TGN, ER, and mitochondria is subject to regulation by the miRNA pathway. MiR-375 is the most abundant miRNA in the pancreatic beta-cell and is a potent regulator of insulin secretion and adaptive proliferation[41–44]. Establishing an association between *Pitpna* and miR-375 suggests a framework for how the beta-cell exerts regulatory control over its critical functions such as granule maturation, exocytosis, and mitochondrial dynamics. We previously demonstrated how miR-375 targets (e.g., *Cadm1*, *Gphn*, *Elavl4*, and *Mtpn*) regulate beta-cell secretion[43,44]. Inclusion of *Pitpna* in this regulon amply illustrates the functional diversity of miRNA-targeted genes that mediate exocytosis. We posit that suppression of these genes by miR-375 provides broad regulatory control over the beta-cell secretory machinery and 'secretome' under normal steady state conditions and this circuit may prevent excess insulin release and safeguards the central nervous system from hypoglycemia[84]. Notably, we observed inhibition of *PITPNA* in ND human islets incurred a reduction of AGO2 expression and this may indicate (1) the beta-cell neutralizes miRNA-mediated gene regulation to counter hyperglycemia during T2D and (2) loss of AGO2 expression in T2D islets may characterize a late phase of the disease where compensatory proliferation of the beta-cell has been compromised[42]. These findings reinforce the notion that the miRNA pathway is a critical component for how cells adapt to changes in their metabolic environment as well as demonstrate how disruption of this pathway renders the beta cell incapable of maintaining a proper homeostatic balance with the ultimate result of diabetic disease[84,85].

Lastly, it is notable that consistent with our single cell and total RNA sequencing analysis from T2D human islets, protein expression of PITPNA was dramatically reduced in each donor sample tested compared to levels in islets of non-diabetic subjects indicating either transcription of the gene is repressed or the protein is degraded during disease progression. While the mechanism(s) that diminish PITPNA expression during human T2D are not known, previous studies did not detect aberrant miR-375 expression in isolated islets of T2D subjects nor has expression of miR-375 been correlated with impaired beta-cell function[86–88]. Therefore, it does not appear that dysregulation of this microRNA mediates *PITPNA* silencing in T2D human islets; however, future studies may assess whether alterations in Ago2 levels or other potential microRNAs targeting this gene occur in beta-cells. Furthermore, defective proglucagon gene expression was observed in the *Pitpna* whole-body knockout mouse indicating the premature lethality of these animals was due to insufficient glucose production to sustain the central nervous system[50]. This observation suggests PITPNA may also facilitate vesicle formation and hormone release from other endocrine cells of the pancreas including alpha and delta cells. Single-cell transcriptomics analysis revealed decreased *PITPNA* levels in delta cells of T2D human islets and reduced somatostatin levels have been previously observed in pancreata of T2D human subjects[89]. As somatostatin is widely described for its inhibitory actions throughout the body, this reduction may constitute a compensatory action of the delta cell population; however, investigation into PITPNA in these other populations of the endocrine pancreas remains to be further studied.

In summary, this study describes several important conceptual advances. These include: (i) establishment of PITPNA as a major regulator of PtdIns-4-P signaling in the TGN of human pancreatic beta-cells, (ii) demonstration that PITPNA is required for efficient insulin granule maturation, docking, secretion, and proinsulin processing in mammalian (including human) pancreatic beta-cells, and (iii) demonstration that restoration of PITPNA expression in human T2D beta-cells rescues insulin secretion, granule maturation and alleviates ER stress. These data not only highlight PITPNA deficiency as a major contributing factor to reduced insulin output and beta-cell failure, but also report a functional crosstalk between the miRNA pathway and lipid signaling control of membrane trafficking factors that are relevant to human diabetes. This study raises the intriguing prospect that enhancing PITPNA expression or activity in islets of T2D human subjects may rescue the multiple defects that contribute to beta-cell degeneration to the extent that physiologically significant activity is revived in the T2D pancreas.

## Methods
All research described in this study complies with all relevant ethical regulations in accordance with the Johns Hopkins University School of Medicine Biosafety Office, the Johns Hopkins Animal Care and Use Committee (ACUC), and the Johns Hopkins Medicine Institutional Review Board (JHM IRB) in addition to the equivalent regulatory bodies at all collaborating institutions as noted throughout this paper.

### Human islets
Human islets from non-diabetic (ND) and type 2 diabetic (T2D) subjects isolated from cadaveric pancreas were obtained from the Integrated Islet Distribution Program (IIDP), the University of Alberta IsletCore, Prodo Laboratories, and the Nordic Network for Clinical Islet Transplantation (Uppsala, Sweden) with permission from the Johns Hopkins Institutional Review Board (IRB00244487). Human islet cells were obtained from de-identified donors and all organ donors provided written informed consent for use of human islets for research. Relevant archival donor information including diabetes status is listed in Supplementary Table 1. Diabetes status was determined from patient records and available hemoglobin A1c (HbA1c) data.

### Animals
Mice were maintained on a 12-h light/dark cycle with ad libitum access to regular chow food (2016 Teklad global 16% Protein diet, Envigo) and the Johns Hopkins Animal Care and Use Committee approved all experimental procedures under protocol MO18C281. Mice were monitored daily by JHACH vivarium staff and were euthanized by

carbon dioxide or cervical dislocation according to the most recent guidance of the American Veterinary Medical Association (AVMA). Results were obtained from mice of both genders and this information is specified in the figure legends. *Pitpna* whole-body knockout mice were previously described[28,50]. *Pitpna*-floxed mice (VAB line) were generated using a *Pitpna*-floxed allele generated by TALEN-based methods and transplaced into C57BL/6 embryonic stem cells by homologous recombination. Details are available upon request. The successfully targeted *Pitpna* allele had a LoxP sequence inserted upstream of exon 8 and a neomycin cassette (flanked by Frt sequences)-LoxP sequence inserted downstream of exon 10. The neomycin cassette flanked by Frt sequences was removed by crossing to an FLP deleter strain. In the resulting strain (i.e., *Pitpna*-floxed strain), exons 8–10 of *Pitpna* were flanked by LoxP sequences. Deletion of exons 8–10 generates a *Pitpna* null allele. The primers for genotyping the *Pitpna*-floxed allele were: TAMU002_LoxP_F: 5′-AGTGAGTTCCAAAA TGGCCAGGTT-3′; and TAMU002_LoxP_R: 5′-GCCAGTTCTTTTTGTCG CTGTGAA-3′. The size of the PCR product was 242 bp for the wild-type *Pitpna* allele, and 312 bp for the floxed *Pitpna* allele. Floxed *Pitpna* mice were crossed with *Ins1-Cre* mice purchased from Jackson Labs[51]. Floxed *Ago2* mice were generated and crossed with *Ins-Cre* mice from P. Herrera as described[42]. Numbers of animals are reported in each figure legends, and experiments were conducted in a blinded manner where the genotype is unknown during actual testing.

## Gene expression analysis in mouse and human islets
Total RNA was extracted using the TRIzol reagent (Invitrogen). Quantitative real-time PCR (qRT-PCR) for *miR-375* was quantified by TaqMan Assays using the TaqMan MicroRNA Reverse Transcription Kit and hsa-miR-375 primer sets (Thermo Fisher Scientific, 000564). *MiR-375* levels were normalized to *miR-U6* expression. For the expression of gene mRNAs, cDNA was synthesized using RevertAid First Strand cDNA synthesis kit (Fermentas), and qRT-PCR was measured using gene-specific primers with FastStart SYBR Green PCR Master Mix (Roche) on a StepOne Real-Time PCR System (ThermoFisher) (Supplementary Tables 2 and 5). Human islet expression data and accompanying donor information were previously published[47] and are publicly accessible at Gene Expression Omnibus (GEO accession number GSE50398). Briefly, RNA-seq datasets were downloaded, trimmed (TrimGalore), and mapped to GRCh38 (HISAT2 mapper)[90]. Read counts for each sample were generated in SeqMonk software and normalized. The expression levels for *PITPNA* and *INSULIN* were correlated to the published clinical data included with the GEO submission. Single-cell RNA-seq data (GEO accession GSE85241)[48] was downloaded from https://hemberg-lab.github.io/scRNA.seq.datasets/human/pancreas/ as a log normalized single cell experiment R object and processed using the R package Seurat v3.2.3[91].

## Analytic procedures
Insulin measurements from plasma and pancreatic extracts were measured by ELISA (Crystal Chem), blood glucose and luciferase assays were measured as described[44]. Islet morphometric analysis was performed on 8 μm sections of paraffin-embedded pancreas approximately 150–200 μm apart. Sections were dewaxed, washed, and stained for insulin (Dako A0564, Agilent IR00261-2), glucagon (Millipore MABN238), Ki-67 (NovaCastra), Pitpna (Proteintech, 16613-1-AP), GM130 (BD, 610822), KDEL (Novus, NBP1-97469), Calreticulin (NOVUS, NBP1-47518), Giantin (NOVUS, AF8159), CD63 (Bio-Rad, MCA4754), or TUNEL (Roche cat. no. 11684795910). Cell numbers from all islets in 3–7 sections were counted with ImageJ software from 20X images obtained using a Nikon A1RSI Spectral Confocal Microscope. In vivo insulin release and glucose (GTT) tolerance tests were performed following a 16-h fast and intraperitoneally injection of glucose (2 g/kg BW). Insulin secretion from isolated islets was performed as described[44]. All antibodies and key reagents used in this study are listed in Supplementary Tables 4 and 5, respectively.

## Cell culture, immunoprecipitation, and western blotting antibodies
**Insulin secretion assay.** MIN6 cells were cultured in DMEM (Invitrogen) containing 4.5 g/L glucose supplemented with 15% v/v heat-inactivated FCS, 50 μM β-mercaptoethanol, and 50 mg/mL penicillin and 100 mg/ml streptomycin and were a kind gift from Paolo Meda (University of Geneva, Switzerland). Cells were preincubated for 30 min in HEPES-balanced Krebs-Ringer bicarbonate buffer (KRH: 119 mM NaCl, 4.74 mM KCl, 2.54 mM $CaCl_2$, 1.19 mM $MgCl_2$, 1.19 mM $KH_2PO_4$, 25 mM $NaHCO_3$, and 10 mM HEPES, pH 7.4) containing 0.5% BSA with 5 mM glucose and then were incubated for 2 h with various concentrations of glucose. Insulin release was measured by ELISA (Crystal Chem, cat. no 90080)[41].

**Western blotting.** The following primary antibodies were used for Western blots at 1:1000 dilution: PITPNA (Abcam, ab180234), Cadm1 (MBL, CM004-3), Gephyrin (BD Biosciences, 610585), CHOP (Cell Signaling, 2895S), BiP/GRP78 (Cell Signaling, 3177S), DRP1 (Proteintech, 12957-1-AP), β-Actin (Cell Signaling, 3700S), and γ-Tubulin (Sigma, T6557). The following primary antibodies were used for immuno-fluorescence: PITPNA (1:200, Sigma, SAB1400211). Antibodies were used on paraffin-embedded pancreata fixed in 4% paraformaldehyde for 3 h. Image densitometry of 16-bit TIF images for all western blots was performed using ImageJ. All original uncropped western blot images are presented in Supplementary Figs. 7–13.

**Transfection of microRNA mimics and related studies.** MicroRNA mimics and siRNA pools were purchased from Qiagen GmbH (Germany) and scrambled pool controls are defined as an equimolar stock solution of either 48 random siRNA sequences, or 12 unique mimics of miRNAs not expressed in the beta-cell (i.e., miR-122, miR-1) and not predicted by the TargetScan algorithm to bind the 3′UTR of *Pitpna*. For biochemical fractionation, an eight-step sucrose gradient was performed on MIN6 cells as described previously[43]. Briefly, MIN6 cells were washed, pelleted, and resuspended in buffer containing 5 mM HEPES, 0.5 mM EGTA, and 1X Complete Protease inhibitors (Roche Applied Science) at pH 7.4 and homogenized. Homogenate was spun at $3000 \times g$ for 10 min at 4 °C, and the post-nuclear supernatant was loaded onto an 8-step discontinuous sucrose density gradient (HEPES-buffered 0.2–2 M sucrose) and centrifuged at $242,000 \times g$ for 2 h at 4 °C using an MLS50 rotor (Beckman Coulter). Extracellular acidification rate (ECAR) and oxygen consumption rate (OCR) were measured in MIN6 cells using an XF24 Analyzer (Seahorse Bioscience, MA, USA).

## Lentiviral-mediated over-expression and knockdown in isolated human islets
Lentiviruses were generated after subcloning the *PITPNA* cDNA sequence into the expression vector pCCL-cPPT-PGK-IRES-WPRE (Addgene). The resulting construct was transfected along with packaging plasmids pMD2.G and pSPAX2 (Addgene) into HEK293T cells (ATCC, CRL-3216). Cell culture media containing the virus was collected 48 and 72 h after transfection, concentrated, and stored at −80 °C. Knockdown of *PITPNA* by MISSION shRNA vectors (Sigma-Aldrich) was confirmed in human pancreatic 1.1B4 cells and isolated islets. Human islets were treated with non-overlapping shRNAs against the human *PITPNA* mRNA (accession number NM_006224), and TRCN00000299703 (SHCLNV 06302009MN) was used for all studies. TRC2 pLKO.5 Lentiviral Transduction Particles (pLKO.5-puro non-Mammalian shRNA Control Plasmid DNA; SHC00204V) were used to treat control human islets. Polybrene (Santa Cruz Biotechnology, Cat# sc-134220, Texas, USA) was added to the media with the final concentration of 10 μg/ml before infection. In

brief, 250 islet equivalents (IEQ) seeded in each 12-well plate were infected with each lentivirus at an M.O.I of 20 for 48–72 h to ensure complete infection. All plasmids and shRNAs related to these procedures are listed in Supplementary Table 3.

## Total internal reflection fluorescence (TIRF) microscopy

For TIRF microscopy experiments, human islets were obtained from the Nordic Network for Clinical Islet Transplantation, Uppsala Sweden, with ethical clearance (Uppsala Regional Ethics Board 2006/348) and the donor families' written informed consent. Islets (donor IDs R442, 2583, 2585) were dispersed into single cells in cell dissociation buffer (Thermo Fisher Scientific) supplemented with trypsin (0.005%, Life Technologies), washed and plated in serum-containing medium on 22-mm polylysine-coated coverslips, allowed to settle overnight, and then transduced with adenovirus coding for the granule marker NPY-tdmOrange2. Lentiviral vectors expressed EGFP which was used to select cells for imaging as described previously[68] using a lens-type total internal reflection (TIRF) microscope, based on an AxioObserver Z1 with a 100x/1.46 objective (Carl Zeiss). TIRF illumination with a decay constant of -100 nm (calculated based on exit angle) was created using two DPSS lasers at 491 and 561 nm (Cobolt, Stockholm, Sweden) that passed through a cleanup filter (zet405/488/561/640x, Chroma) and was controlled with an acousto-optical tunable filter (AA-Opto, France). Excitation and emission light were separated using a beamsplitter (ZT405/488/561/640rpc, Chroma) and the emission light chromatically separated (QuadView, Roper) onto separate areas of an EMCCD camera (QuantEM 512SC, Roper) with a cutoff at 565 nm (565dcxr, Chroma) and emission filters (ET525/50 m and 600/50 m, Chroma). Scaling was 160 nm per pixel. Cells were imaged in (mM) 138 NaCl, 5.6 KCl, 1.2 MgCl$_2$, 2.6 CaCl$_2$, 0.2 diazoxide (to prevent spontaneous depolarizations), 10 D-glucose, 5 HEPES (pH 7.4 with NaOH) at -35 °C. Exocytosis was evoked with high 75 mM K$^+$ (equimolarly replacing Na$^+$), applied by computer-timed local pressure ejection through a pulled glass capillary. Exocytosis events were identified manually based on the characteristic rapid loss of the granule marker fluorescence (1–2 frames).

## Analysis of intracellular calcium

Intracellular calcium [$_i$Ca$^{+2}$] assay was performed (Fluo-4NW Invitrogen, F36206, excitation 494 nm, emission 516 nm) according to the manufacturer's instructions. Briefly, [$_i$Ca$^{+2}$] was recorded for 60–90 s after addition of the KCl. Human islets were fixed in black 96-well optical bottom plates with poly-D-lysine coating. After the dye loading for an hour, the recording was done under confocal microscope (×40 objective) at room temp using an excitation filter of 488 nm. Fold change [$_i$Ca$^{+2}$] was calculated from the baseline fluorescence recorded during the first 5 s before the addition of KCl. Images were captured at 1 s intervals for up to 60 s and the intracellular free calcium concentration is represented by mean fluorescence intensity.

## Immunostaining and confocal microscopy

The following primary antibodies were used for immunofluorescence: anti-GOLPH3 (1:1000, Abcam, ab98023), anti-Golgin97 (1:100, Invitrogen, A-21270), anti-PtdIns-4-P (1:500, Echelon Biosciences cat. no. Z-P004), and anti-insulin (1:1000, Dako cat. no A0564). For immunostaining, both primary and secondary antibodies were diluted in 1x PBS containing 2.5% bovine serum albumin and 0.2% Triton-X-100. Antibody incubation steps (primary antibody: 3–4 h; secondary antibody: 1 h) were performed in a humidified chamber protected from direct light. Alexa Fluor 488, 594, 647 anti-rabbit, anti-mouse, or anti-guinea pig secondary antibodies used in this study are listed in Supplementary Table 4. Cell nuclei were stained with DAPI and mounted with Fluorsave reagent (MilliporeSigma, 345789) for fluorescence microscopy. Confocal images were acquired on a Nikon TiE confocal microscope using the NIS-Elements software with ×60 oil immersion objective.

Images were imported into the *Fiji* version (http://fiji.sc) of the *ImageJ* software and the colocalization analyses were performed using the Coloc2 plugin (https://imagej.net/plugins/coloc-2)–an automated system that evaluates the fluorescent intensities of every pixel within an area of interest. Quantification of colocalization was performed using Pearson's correlation coefficient. The Pearson's correlation coefficient reflects the degree of linear relationship between two variables; in this case, the fluorescence intensities of two fluorescently tagged proteins GOLPH3 and Golgin97.

## Transmission electron microscopy (TEM)

Isolated mouse islets and MIN6 cells were fixed in 2% paraformaldehyde/2.5% glutaraldehyde in 0.1 M Sodium Cacodylate buffer (cat. no. 15960-01 Electron Microscopy Sciences) for 2 h at 4 °C and then stained in 1.0% osmium tetroxide (cat. no. 19100 Electron Microscopy Sciences) for 1 h. After dehydrated in ethanol, cells were embedded with Spurr's Low Viscosity Embedding Kit (cat. no. EM0300-1KT, Electron Microscopy Sciences), sectioned (70–90 nm thick), placed on Formvar (200 mesh) copper grids and contrasted with uranyl acetate (cat. no. 22409 Electron Microscopy Sciences) and lead citrate (cat. no. 22410 Electron Microscopy Sciences). Imaging was performed on a Philips Morgagni transmission electron microscope and acquired images were analyzed with respect to insulin granule and mitochondrial morphology.

## Mass spectrometric analysis

**Analysis of MIN6 cells.** Cells were harvested by trypsinization and washed twice with ice-cold PBS. A modified protocol of Bligh & Dyer was used to extract lipids from cells[92]. Briefly, 900 μL of chloroform:methanol (1:2, v-v) (Thermo Fisher) was added to 2 × 10$^6$ cells. After vortexing for 1 minute and incubating for 15 minutes on ice, 300 μL of chloroform was added to the mixture, followed by mild vortexing and addition of 300 μL distilled water. The mixture was vortexed for 2 min and centrifuged at 20,000 × $g$ for 2 min at 4 °C. The lipids were isolated from the lower organic phase. The sample was vacuum dried (Thermo Savant SPD SpeedVac) and the dried extract resuspended in 200 μL of chloroform:methanol (1:2, v/v) containing standards: PC 28:0, PE 28:0, PI 25:0, PG 28:0, PA 28:0, PS 28:0, LPC 17:0, LPE 14:0, d$_6$-CE 18:0, and d$_5$-TAG 48:0 (Avanti Polar Lipids). Phospholipids and neutral lipids were analyzed on an Agilent 1290 HPLC system coupled with an Agilent Triple Quadrupole mass spectrometer 6460, using Zorbax Eclipse Plus C18 column, 2.1 × 50 mm, 1.8 μm. The mobile phases were: A (acetonitrile:10 mM ammonium formate, 40:60) and B (acetonitrile:10 mM ammonium formate, 90:10). For phospholipids separation the gradient was as follows: start at 20% B, to 60% B in 2 min, to 100% B in 5 min, hold at 100% B for 2 min, back to 20% be in 0.01 min, hold 20% B 1.79 min (total runtime 10.8 min), the flow rate was 0.4 mL/min and the column temperature 30 °C. For neutral lipids separation the gradient was as follows: start at 20% B, to 75% B in 2 min, to 100% B in 4 min, hold at 100% B for 3 min, back to 20% be in 0.01 min, hold 20% B 1.79 min (total runtime 10.8 min), the flow rate was 0.4 mL/min and the column temperature 40 °C. Positive and negative electrospray ionization (ESI) was undertaken using the following parameters: gas temperature, 300 °C; gas flow, 5 l/min; nebulizer, 45 psi; sheath gas temperature, 250 °C; and sheath gas flow, 11 l/min; capillary voltage, 3.5 kV. Phospholipids and neutral lipids were measured using multiple reaction monitoring (MRM). Each biological replicate was measured twice and the average measurement used for analysis. MRM transitions areas were normalized to the areas of the internal standard from the same lipid class. Identification of peaks were based on retention time (RT) and specific MRM transitions for each lipid. Lipid species with CoV >25% were removed. Raw peak areas were integrated using Agilent MassHunter Quantitative Analysis software (version B.06.00), with further analysis performed in Microsoft Excel (v 16.0). Individual lipid species were quantified by comparison

with spiked internal standards and lipid species with Signal-to-Noise <3 were removed. The molar fractions of individual lipid species and each lipid class were normalized to total lipids as follows: individual lipid intensities were divided by the relevant internal standard's intensity and multiplied by the standard's concentration; the obtained concentration value was divided by the sum of all lipids concentrations to yield molar fractions (mol%). Data were plotted using GraphPad Prism version 9.1.0. and R 4.1.0 (package ggplot2 and tidyr). Each biological replicate was measured twice and the average measurement was used for analysis. Comparisons between two groups were evaluated using an unpaired Student's t test with Benjamini-Hochberg correction for mulitiple testing. Data summary for mass spectrometric analysis of MIN6 cells is present in the Source data file.

**Analysis of human islets.** Human pancreatic islets were washed with PBS and collected from 6 well plates then transfer into Eppendorf Lo-Bind polypropylene centrifuge tubes followed by centrifugation at 30,000 × $g$ for 1 min at 4 °C. After removing the PBS, 0.5 M TCA was added to the pellet, vortexed, and incubated on ice for 10 min. The cooled mixture (TCA and islets) was centrifuged at 30,000 × $g$ for 3 min at 4 °C and the supernatant discarded. Finally, 5% (w/v) TCA containing 10 mM EDTA was added to the pellet and vortexed, and then stored at −80 °C[93,94]. Internal standards of 20 ng PtdIns(4,5)P2, 20 ng PtdIns(4)P, and 100 ng PtdIns were added to the precipitates prepared from samples as described above. The lipid analytical internal standards were ammonium salts of, 1-heptadecanoyl-2-(5Z,8Z,11Z,14Z-eicosatetraenoyl)-sn-glycero-3-phospho-(1′-myo-inositol-4′,5′-bisphosphate) [17:0, 20:4 PtdIns(4,5)P2], 1-heptadecanoyl-2-(5Z,8Z,11Z,14Z-eicosatetraenoyl)-sn-glycero-3-phospho-(1′-myo-inositol-4′-phosphate) [17:0, 20:4 PtdIns(4)P], and 1-heptadecanoyl-2-(5Z,8Z,11Z,14Z-eicosatetraenoyl)-sn-glycero-3-phospho-(1′-myo-inositol) [17:0, 20:4 PI] from Avanti Polar Lipids (LIPID MAPS MS Standards)[93]. Following the addition of internal standards, 670 µl of ice-cold chloroform–methanol–12.1 N HCl (40:80:1) was added to each sample, after which the samples were vortexed for 2 min and allowed to sit on ice for another 10 min. Then, 650 µl of chloroform and 300 µl of 1 N HCl were added to generate two phases. Vortexed samples were centrifuged at 10,000 × $g$ for 2 min to generate 2 separate phases. The lower phases were collected in a fresh 2 ml tube. An additional 950 µl of a mixture of chloroform, methanol, 1.74 M HCl (v:v:v) was added to the upper phase followed by vortexing and centrifugation at 10,000 × $g$ for an additional 2 min. The resultant lower phase was combined with the previously collected lower phase and the combination dried under a nitrogen stream using a BiotageTM evaporator. The dried extracts were derivatized (methylated) with TMS-DM and resuspended in methanol for analysis[94]. Samples were resuspended in 25–100 µl 100% methanol (LC-MS Optima grade, Fisher) prior to chromatographic separation at ambient temperature using a C4 column (Waters Acquity UPLC Protein BEH C4, 1.7 µm 1.1 × 100; 300 A). A Waters Acquity FTN autosampler set at 4 °C injected 5 µl of sample via Waters Acquity UPLC. For chromatography of phosphoinositides the mobile phase was delivered over an 18.5 min runtime at a flow rate of 0.1 ml/min by a Waters Acquity UPLC. The gradient was initiated with 10 mM formic acid in water (A)/10 mM formic acid in acetonitrile (37:63 v/v) (B), held for 2 min, then increased to 15:85, v/v in 10 min, then increased to 100% B and held for 2.8 min followed by 3 min re-equilibration to starting conditions. The effluent was monitored by a Waters XEVO TQ-S MS/MS in multiple reaction monitoring mode (MRM) using electrospray and positive ion mode with post column infusion of 50 µM Na formate at 5 µL/min. to encourage the formation of sodiated adducts. Derivatized phosphoinositide species were quantified by targeted analysis. Specifically, peak areas for lipid species and standards were quantified by integrating curves using Waters' MassLynxs software 4.2 employing TargetLynxs for peak integration and outputs. (Waters Corporation, Milford, MA, USA). For absolute calibrations and comparisons of different samples, peak areas of endogenous species were normalized to those of the corresponding internal standards. Two approximate assumptions were made: (1) the extraction and detection efficiency of phosphoinositides with each of the different fatty acyl combinations was the same as for the 37:4 standard, and (2) peak areas were proportional to lipid concentration. Data summary of mass spectrometric analysis of human islets is included in the Source data file.

## Statistics and reproducibility

All results are expressed as mean ± standard error (SEM) and statistical analysis including all n numbers is summarized alongside primary data in the Source data file that accompanies this paper Results with human and mouse pancreatic islets and cell lines were all achieved with test groups of biologically independent samples of $n \geq 3$ to determine reproducibility. Experiments with transgenic mice implemented test groups of each genotype of $n \geq 4$ to determine reproducibility. All animal experiments implemented cohorts derived from multiple breeding pairs and were repeated at least once to determine reproducibility. A P-value of less than or equal to 0.05 was considered statistically significant. *$P < 0.05$, **$P < 0.01$, and ***$P < 0.001$. All graphical and statistical analyses were performed using the Prism8 software (Graphpad Software, USA) and Microsoft Excel. All measurements are from distinct samples and none of the results are derived from repeated measurement of the same sample. Comparisons between datasets with two groups were evaluated using an unpaired Student's t test. ANOVA analysis was performed for comparisons of three or more groups.

## Reporting summary

Further information on research design is available in the Nature Portfolio Reporting Summary linked to this article.

## Data availability

All primary data including original uncropped western blots and raw mass spectrometry data generated in this study are provided in the Supplementary Information/Source data file provided with this paper. The following datasets containing expression analysis from human islet cells were re-analyzed for this study: GEO accession number GSE50398[47] and GEO accession GSE85241[48]. Source data are provided with this paper.

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

## Acknowledgements

The authors thank David Castle, T. Osborne, L. Nagy, and M. Elena-Arango for assistance in the conduct of this work. This work was funded by Johns Hopkins All Children's Hospital (M.N.P.), NIH grants R01 DK135688 (M.N.P.) and R35 GM131804 (V.A.B.), the Helmholtz Gemeinschaft (M.N.P.), two European Foundation for the Study of Diabetes EFSD/Lilly Programme Grants (M.N.P. and S.B.), Swedish Science Council (S.B.), NovoNordisk Foundation (S.B.), and the Deutsche Forschungsgemeinschaft (YA 721/3-1 to X.Y., LU 1455/6-1 to Y.L.). Human pancreatic islets and/or other resources were provided by the NIDDK-funded Integrated Islet Distribution Program (IIDP) (RRID:SCR _014387) at City of Hope, NIH Grant # U24DK098085 and the JDRF-funded IIDP Islet Award Initiative BS522P (to M.N.P.).

## Author contributions

Y.T.Y. and C.S. performed the primary expression analysis, animal husbandry, electron microscopy image analysis, immunohistochemical, and morphometric analysis and edited the manuscript. X.Y., Y.W., S.J., S.K., S.N., and A.A., performed expression analysis. L.L. and S.B. performed TIRF microscopy and edited the manuscript. Y.L., A.P., and M.M. performed immunohistochemical analysis. A.G. and F.v.M. reanalyzed public expression datasets. Y.W., A.C.-G., and M.W. performed and analyzed the lipidomic analysis. A.T.K. quantified phosphoinositides. P.A. edited the manuscript. Z.X. and V.A.B. developed and provided the Pitpna mutant animal lines and edited the manuscript. M.N.P. conceived and designed the study, wrote the manuscript, and is the guarantor of this work and takes responsibility for the integrity of the data and the accuracy of the data analysis. All authors contributed to interpretation of the data and approved the final version of this manuscript.

## Competing interests

The authors declare no competing interests.
