## [Peer Review File · Nature Communications]

REVIEWER COMMENTS

Reviewer #1 (Remarks to the Author):

The authors demonstrate that PITPNA dependent PtdIns-4-P synthesis has an expected role to integrate the function of membrane compartments in pancreatic beta cells that are important for GSIS and other related beta cell functions—including insulin granule biogenesis and maturation, proinsulin processing, and resistance to ER stress and progressive apoptosis. Importantly, the function of PITPNA is generally similar in MIN6 cells, mouse islets/beta cells, and human islets/beta cells.

My general feeling is that the data and analysis are exhaustive and detailed, and I generally have no question about what has been assembled here; however, the results section is difficult to read owing to the excessive detail. It seems the section could be improved considerably with some careful editing.

The effects of over expression or knock down of PITPNA are sometime small. For example, although the entire curve is mildly reduced, first phase insulin secretion is still intact in beta cell conditional KO mice (See Fig 2c). This seems unexpected if PITPNA is so central to beta cell function. I wonder how islets continue to function after PITPNA knockout? How do the islets get around the absence of such an important enzyme, unless other unknown mechanisms also play important roles?

The analysis of PITPNA in human islets focuses our understanding of possible defects underlying T2D, which provide important translational results. The story is certainly written in a way to emphasize the critical importance of PIPPNA in T2D. Regardless, it is difficult to know how PITPNA dysregulation contributes to the progression of T2D, especially whether it is the driving mechanism. Genetic/viral strategies nicely show that increased PITPNA improves GSIS in human islets/beta cells while decreased PITPNA impairs function. Regardless, the mechanism of PITPNA failure in T2D beta cells is unclear. This is an important weakness. Even with all the data, it is impossible to establish whether the loss of PITPNA is a cause of beta cell failure or one of the consequences of beta cell failure arising from the increased demand upon beta cells caused by hepatic and peripheral insulin resistance. At least the authors show that increased PITPNA cause enhancement of beta cell function in islets from T2D patients.

A mechanism of PITPNA failure in beta cells of T2D is conspicuously missing. This is disappointing. We are led to think it might be through dysregulated expression of miR-375 that suppresses PITPNA along with other genes. But the evidence is weak or absent in this regard. So, what is the mechanism of dysregulated PITPNA in human islets. Unfortunately, this important question is left open.

Reviewer #2 (Remarks to the Author):

This is a comprehensive and elegant report on functional characterization of phosphatidylinositol transfer protein alpha (PITPNA) as a contributor to beta cell failure in T2D. PITPNA stimulates production of phosphatidylinositol-4-Phosphate (PI-4-P) via PI-4-OH kinase, promoting insulin granule formation.

Through various studies in murine and human islets, the authors were able to demonstrate that ablation of PITPNA associates with ER stress, mitochondrial dysfunction, reduced beta cell mass, hyperglycemia, and reduced glucose-stimulated insulin secretion. Overexpression of PITPNA in human islets from T2D patients rescues the beta cell functionality. Together, the study does suggest that PITPNA might be an attractive target for T2D. One should keep in mind that PITPNA is widely expressed in the body, particularly GI tract and kidney (Human Protein Atlas), thus given key role of this protein in regulation lipid metabolism, targeting it may need to be tissue specific.

Targeted analysis of PIs and PIPs was performed using established protocol, convincingly showing that ablation or overexpression of PITPNA decreases or increases levels of PIPs, respectively, in line with the known biochemistry of this protein. According to the methods, the PIPs were normalized by total PI level. Minor comment: it seems total PIP levels are roughly at the same concentration as total PIs. What's the rationale for using total PI, and not e.g. total protein level?

Reviewer #3 (Remarks to the Author):

In their manuscript the authors report the consequences of global and beta-cell specific knockout of PITPNA in mice. They show hypoglycaemia, ER stress and altered insulin processing. They also show that PITPNA is reduced in T2D patients and that knock-down of the protein in human islets leads to a similar phenotype as in mice. Overexpression of PITPNA in T2D islets elevates islet functions.

Overall, the experiments of this manuscript are well designed and the authors aimed to match mouse, cell line and human data. However, especially for the human data not always the right controls could be provided and some of the results appear to be over-statements.

Figure 1:

a: It would be great to get high res images and to check the colocalization of PITPNA with markers for Golgi, ER and granules. This can be done in mouse islets.

Extended Data Figure 3:

In e the authors state that the experiment was done in mouse islets but the rest of the figure deals with Min6 data. Is there a mistake in the labeling?

Also in Figure e the control values seem to be very low and the cells don't secrete properly. Especially if we compare it to the secretion data in a.

Figure 2:

In 2 h the authors show ultrastructural changes in the distribution of granule types. It would be great if they could also compare the overall density of all granule types.

In general the authors show also for the human islets high quality images and a thorough quantification. It would be great if they could check in these images also the Golgi and ER morphologies and address possible changes in the PITPNA KO. This is especially interesting when thinking about the data presented in Figure 3.

Figure 3:

Concerning the ER stress it would be very interesting if the authors could check their EM images for altered ER and Golgi morphometry. Is the ER swollen compared to controls?

I find their interpretation that granule phenotype feedbacks to ER a bit far-fetched. To me it seems that the problem starts with the Golgi (maybe even at ER level) and the problems with insulin processing are a consequence of that.

Extended Data Figure 4: Why do the authors show OCR only of the Min6? Would it be possible to do these experiments with the conditional KO mice?

Concerning the EM: The authors later discuss a possible impairment of autophagy. Do the authors see altered numbers or ultrastructure of degradative compartments in their EM images? They could also check changes in LC3 levels in the islets.

Figure 4: a-c The problem with these data is that we don't know how well the islets were secreting before the KD experiment. I'm not sure what the n numbers refer to in this experiment. In the bottom of the figure legend the authors state that the data were generated from 3 donors. How come that the n numbers are sometimes higher than 3? Is this referring to individual islets? How efficient was the KD and OE in the individual samples?

d-h: For the KD and OE: How can the authors be sure to image transfected cells? Did the authors include GFP here? If not they likely image a mix of transfected and non-transfected cells. In f and g I cannot see biphasic secretion. Could the scaling of the graph be changed to see it?

Figure 5: Here it would be great to look closer at ER morphometry. I would also like to have more details on the analysis. Were whole images analyzed and the granule numbers normalized for cytoplasm area or were crops made? In a second panel it seems that the cell in the center has very few granules but the authors show a crop with a region with more granules. In this paragraph from line 312 the authors make claims that rather belong in the discussion.

Figure 7:

In b and c it is not clear how the T2D and T2D-OE data relate to ND data. Do the authors have these data and can put them there as a comparison?

A major problem of these experiments is that T2D is a heterogeneous disease. When the authors show the improvement of beta cell function after PIP3NA OE we cannot be sure how good or bad the performance of these samples was before the over expression. Ideally, one should measure insulin release of the T2D islets before OE and after. Could these experiments be done with a mouse model for T2D?

Also, the OE partially improves granule density and docked vesicles, but not to the values of ND samples. The authors should make this distinction and tone down the interpretation of the data a little bit. Also here it would be great to know what the n refers to (number of donors?).

Discussion:

I agree with the authors that PIP3NA plays an important role in granule formation and maturation and to me the data indicate that if the protein is knocked-down problems in insulin processing at the Golgi and possibly ER result in the observed phenotype. I don't think that PIP3NA OE fully rescues beta cell function and I would recommend to use terms like alleviate.

Reviewer #4 (Remarks to the Author):

Yeh et al. characterize the important function of Phosphatidylinositol transfer protein alpha (PIP3NA) in PtdIns-4-P signaling in beta-cells and identified decreased PIP3NA in isolated islets in T2D which leads to defects in insulin processing and granule maturation, beta-cell dysfunction, ER stress and apoptosis, all which could be reversed by restoration of PIP3NA in T2D islets of T2D.

The paper is novel and experiments sound. Results are based on a beta-cell specific PIP3NA-KO mouse model, a mouse beta-cell line and human islets. I only have some minor comments:

1) The order in the 1st chapter is a bit confusing, as there are only Suppl Figures connected to it; as figures are non restricted, Suppl.Fig.1a-d should be moved into Fig.1 and then continued.

“ is expressed in pancreatic beta-cells (Figure 1a)” should be changed to “in islets and not restricted to beta-cells...”

Labelling should be consistent and diseased organ donors not called “healthy”-change to “non” or better “control” for all.

It should be made clear that 1b-d is a data reanalysis of Ref.49

2) Ins-Cre, Pitpna-flox/flox mice have some impaired glucose tolerance and also lower insulin. However, it seems that glucose stimulated insulin secretion in vivo is only marginally or unchanged compared to WT mice. I am not so convinced by the data that there is necessarily a defect in glucose stimulated insulin secretion in the Pitpna-depleted beta-cells, as mice have already lower basal insulin secretion.

Please show the stimulatory index at 15 min over 0 min before glucose injection. Error bars are missing for the in vivo GSIS. The stats is for the AUC analysis or is it for each data point. Please add explanation for the P-values.

3) TUNEL staining looks very unusual with labels accumulated in the cytosol. TUNEL as a fragmented DNA marker should mainly and very clearly be localised in the nucleus, the provided view looks rather like an unspecific background staining. Please re-evaluate the staining by using + and -controls.

Also in Suppl.F.2h, TUNEL stain isn't convincing, the error points to a TUNEL- cell (or covers another cell?), while the other isn't really ins+?

Beta-cell mass analysis is missing in the methods-

4) Fig.7b shows insulin release from isolated islets from T2D donors after PITPNA overexpression. I guess data are shown after glucose stimulation (5/15 min), please add to legend.

Fig.7a: what is total protein? Is this an unspecific band on the blot? Please explain.

The results from human T2D islets are important but quite limited. Did re-expression of PITPNA also affect ER stress and survival as important functions of PITPNA?

RESPONSE TO REVIEWERS

REVIEWER COMMENTS

Reviewer #1 (Remarks to the Author):

The authors demonstrate that PIPNA dependent PtdIns-4-P synthesis has an expected role to integrate the function of membrane compartments in pancreatic beta cells that are important for GSIS and other related beta cell functions—including insulin granule biogenesis and maturation, proinsulin processing, and resistance to ER stress and progressive apoptosis. Importantly, the function of PIPNA is generally similar in MIN6 cells, mouse islets/beta cells, and human islets/beta cells. My general feeling is that the data and analysis are exhaustive and detailed, and I generally have no question about what has been assembled here; however, the results section is difficult to read owing to the excessive detail. It seems the section could be improved considerably with some careful editing.

We thank Reviewer 1 for their supportive comments and the Results section has been edited to remove extraneous wording and to improve its clarity.

The effects of over expression or knock down of PIPNA are sometime small. For example, although the entire curve is mildly reduced, first phase insulin secretion is still intact in beta cell conditional KO mice (See Fig 2c). We agree with Reviewer 1's assessment that loss of PIPNA in beta-cells (as depicted in experiments implementing the conditional beta-cell-specific knockout mouse or in treated isolated islets) does not completely abolish glucose-stimulated insulin secretion (GSIS). We would however highlight that the observed difference in GSIS in *Ins-Cre, Pitpna^{flox/flox}* mice is measured under an acute time frame (15 minutes), and that more importantly, *Ins-Cre, Pitpna^{flox/flox}* mice are hyperglycemic under random-fed steady-state conditions reflecting that under chronic conditions, PIPNA is a physiologically-relevant gene in beta-cell function.

This seems unexpected if PIPNA is so central to beta cell function. I wonder how islets continue to function after PIPNA knockout? How do the islets get around the absence of such an important enzyme, unless other unknown mechanisms also play important roles?

The precise reason for the absence of a more severe phenotype is unknown to us at this time. We agree that "other unknown mechanisms also play important roles". One potential factor is a related family member, PIPNB, also mediates PtdIns-4-P-dependent recruitment of GOLPH3 to Golgi membranes and has previously been shown to act in redundant fashion to regulate development of the neocortex (Xie et al Dev Cell 2018). We believe that beta-cells (at least in the short term) can accommodate loss of PIPNA due to this built-in redundancy and we believe that many critical pathways have evolved to incorporate functional redundancy/overlap to avert a complete loss of function (i.e. cessation of insulin release) that leads to disease. However, based on the hyperglycemia observed in *Ins-Cre, Pitpna^{flox/flox}* mice, we hypothesize that redundant pathways cannot fully compensate for the chronic absence of PIPNA in beta-cells ultimately leading to beta-cell failure and hyperglycemia.

The analysis of PIPNA in human islets focuses our understanding of possible defects underlying T2D, which provide important translational results. The story is certainly written in a way to emphasize the critical importance of PIPNA in T2D. Regardless, it is difficult to know how PIPNA dysregulation contributes to the progression of T2D, especially whether it is the driving mechanism. Genetic/viral strategies nicely show that increased PIPNA improves GSIS in human islets/beta cells while decreased PIPNA impairs function. Regardless, the mechanism of PIPNA failure in T2D beta cells is unclear. This is an important weakness. Even with all the data, it is impossible to establish whether the loss of PIPNA is a cause of beta cell failure or one of the consequences of beta cell failure arising from the increased demand upon beta cells caused by hepatic and peripheral insulin resistance. At least the authors show that increased PIPNA cause enhancement of beta cell function in islets from T2D patients.

We would respectfully disagree with the statement “it is impossible to establish whether the loss of PIP3 is a cause of beta cell failure or one of the consequences of beta cell failure”. In this study, we present our findings showing 1) beta-cell specific knockout mouse of *Pitpna* results in impaired GSIS, reduced beta-cell mass and random-fed hyperglycemia and 2) PIP3 is significantly reduced in T2D human islets. Together we would argue these observations point to a degree of causality in human beta-cell failure.

To be clear, we believe that beta-cell failure is likely not a singular event or caused by a single factor (DNA, RNA or protein). This is based on the published work showing reduced beta-cell expression of numerous genes in T2D human islets and we would argue it is virtually impossible with current technologies to lay claim to identifying ‘the sole cause for human beta-cell failure’. There may be hundreds of functionally-relevant genes down-regulated in a manner similar to PIP3, and therefore, how likely is it that one factor preceded all of the other ‘consequences’? As discussed above, compensatory mechanisms enable beta-cells to respond to increased demand for insulin and after years of chronic duress, beta-cell failure ensues. Given that the full repertoire of functionally-relevant genes in the beta-cell is currently not fully known, as well as which of those genes are perturbed during human T2D is also not known, we acknowledge that diminished PIP3 is not likely the sole cause of beta-cell failure, but also that a ‘sole cause’ may not even exist and that it is not possible to ascertain with current technology. As Reviewer 1 notes, importantly, we show after restoring PIP3 expression in T2D islets to normal levels, we demonstrate that 1) glucose-stimulated insulin secretion is rescued in addition to insulin granule maturation and 2) expression of many ER stress proteins (i.e. CHOP, BIP, and PDI) is reversed. While this is not evidence for loss of PIP3 as a ‘sole cause of beta-cell failure’, this demonstrates that restoration of PIP3 is adequate to restore GSIS in T2D beta-cells and that loss of PIP3 expression contributes to reducing the secretory output of beta-cells.

A mechanism of PIP3 failure in beta cells of T2D is conspicuously missing. This is disappointing. We are led to think it might be through dysregulated expression of miR-375 that suppresses PIP3 along with other genes. But the evidence is weak or absent in this regard. So, what is the mechanism of dysregulated PIP3 in human islets. Unfortunately, this important question is left open.

To address whether suppression of PIP3 in T2D islets is mediated by the microRNA pathway (i.e. miR-375), we performed western blotting on ARGONAUTE2 (AGO2), an RNA-binding protein and key mediator of microRNA-induced gene silencing including miR-375 (Tattikota et al. Cell Metabolism, 2014). Interestingly, similar to PIP3, expression of AGO2 is dramatically reduced in T2D human islets (**new Figure 1i**). In our previous work, we showed beta-cell specific deletion of *Ago2* in mice impaired compensatory beta-cell proliferation and this indicated that microRNA-mediated gene silencing contributed to the adaptive proliferation of beta-cells in response to increased insulin demand. From this, we hypothesized that multiple microRNAs expressed in beta-cells including miR-375 suppress the secretory machinery to accommodate the cellular energy requirements for proliferation. Therefore, our new observation showing loss of AGO2 expression in T2D human islets may reflect the diminished capacity for compensatory proliferation in beta-cells during T2D. Furthermore, alluding to the discussion above, we tested whether loss of PIP3 in human islets affects AGO2 expression. We measured AGO2 expression by western blotting after inhibition of *PIP3* in human islets and observed reduced AGO2 protein levels (**new Extended Data Figure 1h**). Based on this result, we hypothesize that loss of AGO2 is a consequence of loss of PIP3 in human islets, as loss of *Ago2* in beta-cells of mice (*Ins-Cre, Pitpna^{fl/fl}* mice) was shown to prompt increased expression of genes contributing to secretion including *Pitpna* (Extended Data Figure 1e). The observation that loss of PIP3 coincides with loss of AGO2 in T2D human islets indicates the silencing AGO2 is inadequate to restore expression of PIP3 and can also reflect the diminished capacity for increasing secretion and adaptive beta-cell proliferation to meet the increased insulin demand. Lastly, a previous study has performed miRNA profiling using human T2D islets and noted differential expression of only 2 miRNAs (miR-187 and miR-345) (Locke et al, 2014 Diabetologia). Therefore, from both our own and published observations, reduced PIP3 expression in T2D human islets does not appear to result from increased activity of miRNAs including miR-375.

In summary, we acknowledge the precise mechanism of reduced PIP3 during T2D remains unknown to us and while this is a limitation of our current study, we now show that loss of PIP3 in T2D human islets coincides with loss of AGO2 expression, the RNA-binding protein we previously demonstrated to mediate compensatory beta-cell proliferation. We also demonstrate that inhibition of PIP3 in human islets reduces AGO2 expression indicating loss of AGO2 follows loss of PIP3. These results may suggest that loss of compensatory beta-cell proliferation is a consequence of reduced granule formation that characterizes loss of PIP3 and together these events incur the ER stress that is associated with beta-cell failure during T2D.

Reviewer #2 (Remarks to the Author):

This is a comprehensive and elegant report on functional characterization of phosphatidylinositol transfer protein alpha (PITPNA) as a contributor to beta cell failure in T2D. PITPNA stimulates production of phosphatidylinositol-4-Phosphate (PI-4-P) via PI-4-OH kinase, promoting insulin granule formation. Through various studies in murine and human islets, the authors were able to demonstrate that ablation of PITPNA associates with ER stress, mitochondrial dysfunction, reduced beta cell mass, hyperglycemia, and reduced glucose-stimulated insulin secretion. Overexpression of PITPNA in human islets from T2D patients rescues the beta cell functionality. Together, the study does suggest that PITPNA might be an attractive target for T2D. One should keep in mind that PITPNA is widely expressed in the body, particularly GI tract and kidney (Human Protein Atlas), thus given key role of this protein in regulation lipid metabolism, targeting it may need to be tissue specific. Targeted analysis of PIs and PIPs was performed using established protocol, convincingly showing that ablation or overexpression of PITPNA decreases or increases levels of PIPs, respectively, in line with the known biochemistry of this protein. According to the methods, the PIPs were normalized by total PI level. Minor: it seems total PIP levels are roughly at the same concentration as total PIs. What's the rationale for using total PI, and not e.g. total protein level?

We thank Reviewer 2 for their supportive comments. In brief, there is no perfect molecule for normalization of phosphoinositides. In general, the more means of normalization, the better; however, it should be noted that it is difficult to obtain consistent uniform fractions of cell material to perform protein determination after precipitation. Taking aliquots from cell suspensions is also often problematic depending on the need for extra procedures and time to create uniform cell suspensions. Despite this, it is easier to obtain uniform aliquots from lipid extracts. There will always be the chance that another lipid will also change with any given treatment under investigation; however, based on our experience, the percent change in PI is typically much less than changes in PIP and PIP2. Since PIP is derived from PI, the ratio of the 2 also provides more relevant information. In summary, all these factors need to be taken into consideration when interpreting the data.

Reviewer #3 (Remarks to the Author):

In their manuscript the authors report the consequences of global and beta-cell specific knockout of PITPNA in mice. They show hypoglycaemia, ER stress and altered insulin processing. They also show that PITPNA is reduced in T2D patients and that knock-down of the protein in human islets leads to a similar phenotype as in mice. Overexpression of PITPNA in T2D islets elevates islet functions. Overall, the experiments of this manuscript are well designed and the authors aimed to match mouse, cell line and human data. However, especially for the human data not always the right controls could be provided and some of the results appear to be over-statements.

We thank Reviewer 3 for their constructive remarks and we have performed several new experiments and edited the manuscript as recommended.

Figure 1: a: It would be great to get high res images and to check the colocalization of PITPNA with markers for Golgi, ER and granules. This can be done in mouse islets.

In the revised manuscript, we now provide high resolution image analysis of human islets (**new Figure 1a**) that shows visualization of PITPNA with insulin and markers for the ER (Calreticulin), Golgi (Giantin), and intracellular granules (CD63).

Extended Data Figure 3: In e the authors state that the experiment was done in mouse islets but the rest of the figure deals with Min6 data. Is there a mistake in the labeling?

Reviewer 3 is correct and the result presented in Extended Data Figure 3 was performed in MIN6 cells. The Figure Legend has been corrected to reflect this change.

Also in Figure e the control values seem to be very low and the cells don't secrete properly. Especially if we compare it to the secretion data in a.

The experiment has been re-performed and the new result is shown in Panel e.

Figure 2: In 2 h the authors show ultrastructural changes in the distribution of granule types. It would be great if they could also compare the overall density of all granule types.

Granule density has been quantified in islets of *Ins-Cre*, *Pitpna^{fl/lox/f}* and littermate control mice and this result is presented in new Figure 2h.

In general the authors show also for the human islets high quality images and a thorough quantification. It would be great if they could check in these images also the Golgi and ER morphologies and address possible changes in the PIP2A KO. This is especially interesting when thinking about the data presented in Figure 3. Both ER and Golgi morphologies in islets of *Ins-Cre*, *Pitpna*^{flox/flox} and littermate control mice have been evaluated by electron microscopy and this result is presented in **new Figure 2j**. This analysis reveals loss of *Pitpna* expression in the beta-cell disrupts the morphology of both organelles.

Figure 3: Concerning the ER stress it would be very interesting if the authors could check their EM images for altered ER and Golgi morphometry. Is the ER swollen compared to controls?

As mentioned above, both ER and Golgi morphologies in islets of *Ins-Cre*, *Pitpna*^{flox/flox} and littermate control mice have been evaluated by electron microscopy and this result is presented in **new Figure 2j**.

I find their interpretation that granule phenotype feedbacks to ER a bit far-fetched. To me it seems that the problem starts with the Golgi (maybe even at ER level) and the problems with insulin processing are a consequence of that.

We appreciate this feedback from Reviewer 3 and the text at line 231 was removed. In brief, we believe the primary defect resulting from loss of PIP2A is the reduction in PI4-P synthesis and this in turn reduces granule formation and maturation. We therefore hypothesize that reducing granule production and exocytosis incurs the accumulation of proinsulin that we observed in Figure 5f. Since the precise cause of the ER stress is not known, we would agree that PIP2A-deficiency may incur defects in multiple organelles and have incorporated these hypotheses in the Discussion section (lines 415-418).

Extended Data Figure 4: Why do the authors show OCR only of the Min6? Would it be possible to do these experiments with the conditional KO mice?

OCR measurements in mouse islets were performed; however, maintaining equal numbers of mouse islets throughout the assay was technically challenging and resulted in high variability. Therefore, to follow-up on our observations on OCR, our goal is to now perform stable isotope-resolved metabolomics (SIRM) in a human cell model. This method would facilitate a state-of-the-art assessment of the impact of PIP2A on mitochondrial function.

Concerning the EM: The authors later discuss a possible impairment of autophagy. Do the authors see altered numbers or ultrastructure of degradative compartments in their EM images? They could also check changes in LC3 levels in the islets.

We have assessed LC3 levels in human islets after shRNA-mediated knockdown of PIP2A and observe no change (**Extended Data Figure 1g**).

Figure 4: a-c The problem with these data is that we don't know how well the islets were secreting before the KD experiment. I'm not sure what the n numbers refer to in this experiment. In the bottom of the figure legend the authors state that the data were generated from 3 donors. How come that the n numbers are sometimes higher than 3? Is this referring to individual islets? How efficient was the KD and OE in the individual samples?

To provide clarification, each 'n' refers to each unique islet batch from a single human donor and therefore, n=4 refers to experiments performed on islets from four distinct human donors identified in Supplementary Table 1. In addition, results from human islets shown in Figure 4b and 4c (using islets from 4 and 5 donors, respectively) are from different human donors than those used in Figure 4h (using islets from 3 donors). Throughout our study, efficiency of KD ranged between ~50-80% decrease in PIP2A expression and the efficiency of over-expression was ~1.7 to 4.4-fold increase in PIP2A expression using western blotting.

d-h: For the KD and OE: How can the authors be sure to image transfected cells? Did the authors include GFP here? If not they likely image a mix of transfected and non-transfected cells. In f and g I cannot see biphasic secretion. Could the scaling of the graph be changed to see it?

ShRNA-mediated knockdown and control constructs both expressed EGFP, which was used to select cells and this information has now been added to the accompanying figure legend as well as the Methods section. With regard to Figures 4f and 4g, we provide graphs showing exocytosis binned every 2 s (**new Figure to Reviewers 1a and b**), rather than cumulatively as in Figure 4f and g. The biphasic secretion pattern is more apparent in the overexpression experiment. Changes in scaling would not help to see the biphasic time course (one would have to distinguish between single and double exponential decay by eye).

Figure 5: Here it would be great to look closer at ER morphometry. I would also like to have more details on the analysis. Were whole images analyzed and the granule numbers normalized for cytoplasm area or were crops made? In a second panel it seems that the cell in the center has very few granules but the authors show a crop with a region with more granules. In this paragraph from line 312 the authors make claims that rather belong in the discussion.

ER morphology has been assessed in human islets after over-expressing and inhibition of PIP3 by TEM and these results appear in **new Extended Data Figures 5h and 5i**. The analysis of granule number was performed on whole islet images and the granule number was normalized for cytoplasm area. The two sentences beginning on line 312 have been removed from the Results section.

Figure 7: In b and c it is not clear how the T2D and T2D-OE data relate to ND data. Do the authors have these data and can put them there as a comparison? A major problem of these experiments is that T2D is a heterogeneous disease. When the authors show the improvement of beta cell function after PIP3 OE we cannot be sure how good or bad the performance of these samples was before the over expression. Ideally, one should measure insulin release of the T2D islets before OE and after. Could these experiments be done with a mouse model for T2D?

We appreciate this point from Reviewer 3 and in short, we have GSIS data from ~8-10 unique human donors that contributed to the results in Figures 4a and 4b. As in Figure 4b, we achieve ~2-fold induction of GSIS in control-treated human islets, while in Figure 7, induction appears to be ~1.5-fold in control-treated islets from T2D donors (after 5-minute stimulation). We concur that variability is to be expected in experimentation with T2D human islets and we would add this extends to all primary human tissue. In sharing this concern, we presented data from individual T2D human islets in Extended Data Figure 6. Here we show the impact of PIP3 over-expression on GSIS in islets from 6 unique T2D human donors and results from one donor, HP-22044-01T2D, does not show significantly enhanced secretion after over-expression. However, over-expression of PIP3 in islets of this particular T2D donor was adequate to lower protein expression of CHOP and BIP. This particular example illustrates the experimental variability and led us to presenting all the data from individual T2D donors.

To the point of relating data from T2D donors to ND donors, as described above, the GSIS data from ND donors is presented in Figure 4b. Here we felt the most critical information to be presented is the effect of restoring PIP3 expression in T2D islets on GSIS (as shown in Figure 7b). This experiment as designed is easier to interpret since islets (T2D and T2D-PIP3 OE) are from the same donor with identical basal levels of PIP3 and capacity for GSIS. We felt duplicating Figure 4b next to Figure 7b here was unnecessary since it is already widely accepted that either secretory output of insulin and/or beta-cell mass is reduced in T2D. To the point of measuring GSIS in T2D human islets before and after PIP3 over-expression, this experimentation is not done before over-expression for several reasons: 1) beta-cell function in primary cells does not improve over time since cells are ex vivo in culture, therefore, it is not clear what information is gained, and 2) only limited quantities of T2D islets are available to the research community and this would double the number of islets required.

Lastly, since over-expression of PIP3 in T2D islets improved secretory performance and reverted expression of ER stress proteins, it is unclear what conceptual advance would be achieved by experimenting in a mouse model. Ultimately, our main conclusion is that restoration of PIP3 in T2D human islets reverses beta-cell dysfunction including enhancing GSIS. Hence, we felt the most critical point to convey was how islets from each T2D donor responded to over-expression of PIP3.

Also, the OE partially improves granule density and docked vesicles, but not to the values of ND samples. The authors should make this distinction and tone down the interpretation of the data a little bit. Also here it would be great to know what the n refers to (number of donors?).

We have toned down the interpretation of this result as recommended. N refers to number of unique donors (i.e. n=4 indicates islets are from 4 unique human donors identified in Supplementary Table 1).

Discussion: I agree with the authors that PIP3 plays an important role in granule formation and maturation and to me the data indicate that if the protein is knocked-down problems in insulin processing at the Golgi and possibly ER result in the observed phenotype. I don't think that PIP3 OE fully rescues beta cell function and I would recommend to use terms like alleviate.

We have edited the text throughout the manuscript to minimize the use of the word 'rescue' and have replaced it with 'restore', 'improve', and 'alleviate'.

Reviewer #4 (Remarks to the Author):

Yeh et al. characterize the important function of Phosphatidylinositol transfer protein alpha (PITPNA) in PtdIns-4-P signaling in beta-cells and identified decreased PITPNA in isolated islets in T2D which leads to defects in insulin processing and granule maturation, beta-cell dysfunction, ER stress and apoptosis, all which could be reversed by restoration of PITPNA in T2D islets of T2D. The paper is novel and experiments sound. Results are based on a beta-cell specific PITPNA-KO mouse model, a mouse beta-cell line and human islets.

We thank Reviewer 4 for their comments and have edited the text according to their recommendations.

I only have some minor comments:

1) The order in the 1st chapter is a bit confusing, as there are only Suppl Figures connected to it; as figures are non restricted, Suppl.Fig.1a-d should be moved into Fig.1 and then continued. "is expressed in pancreatic beta-cells (Figure 1a)" should be changed to "in islets and not restricted to beta-cells...Labelling should be consistent and diseased organ donors not called "healthy"-change to "non" or better "control" for all. It should be made clear that 1b-d is a data reanalysis of Ref.49

To accommodate comments from Reviewer 3, Figure 1 has been expanded to include new image analysis highlighting colocalization of PITPNA with subcellular markers for the ER, Golgi, and granules; therefore, due to space constraints, the placement of the data in Extended Data Figure 1 remains unchanged. On line 147, the text regarding the reanalysis of Ref.49 has been edited to state "Datasets obtained from published transcriptomic RNA sequencing analyses..." The text relevant to Figure 1a and the legend for Figure 1d has been changed to Reviewer 4's recommendation.

2) Ins-Cre, Pitpna-flox/flox mice have some impaired glucose tolerance and also lower insulin. However, it seems that glucose stimulated insulin secretion in vivo is only marginally or unchanged compared to WT mice. I am not so convinced by the data that there is necessarily a defect in glucose stimulated insulin secretion in the Pitpna-depleted beta-cells, as mice have already lower basal insulin secretion. Please show the stimulatory index at 15 min over 0 min before glucose injection. Error bars are missing for the in vivo GSIS. The stats is for the AUC analysis or is it for each data point. Please add explanation for the P-values.

Stimulatory index at 15 minutes is now included in new Figure 2c and error bars have also been added to Figure 2c. The statistics refers to the ANOVA analysis and an explanation for the P-value has been added to the figure legend.

3) TUNEL staining looks very unusual with labels accumulated in the cytosol. TUNEL as a fragmented DNA marker should mainly and very clearly be localised in the nucleus, the provided view looks rather like an unspecific background staining. Please re-evaluate the staining by using + and -controls. Also in Suppl.F.2h, TUNEL stain isn't convincing, the error points to a TUNEL- cell (or covers another cell?), while the other isn't really ins+? Beta-cell mass analysis is missing in the methods-

The TUNEL staining has been re-performed and new images have supplanted the previous results. The Methods section has been revised to include beta-cell mass quantification.

4) Fig.7b shows insulin release from isolated islets from T2D donors after PITPNA overexpression. I guess data are shown after glucose stimulation (5/15 min), please add to legend.

The figure legend has been revised according to Reviewer 4's recommendation.

Fig.7a: what is total protein? Is this an unspecific band on the blot? Please explain.

The image of the total protein band conveys a non-specific band on the blot. We have replaced this panel with a new western blot showing actin expression.

The results from human T2D islets are important but quite limited. Did re-expression of PITPNA also affect ER stress and survival as important functions of PITPNA?

In brief, restoration of PITPNA in T2D human islets is adequate to improve GSIS as well as significantly revert expression of ER stress proteins including CHOP, PDI, and BIP (Extended Data Figure 6c).

REVIEWER COMMENTS

Reviewer #1 (Remarks to the Author):

See my comments in red in the attached PDF.

In general, the responses to my original questions seem incomplete, poorly focused on the issue raised, or raise new questions. Specific points are added in RED text to the authors comments (See the attached PFD). I think the authors should add some of their responses into the manuscript. If this was already done, I had trouble finding the text inserts. It would be VERY helpful if the authors could point to where any new results or discussion text might be located in the revised manuscript.

Reviewer #2 (Remarks to the Author):

The authors have adequately addressed my concerns. No further comments.

Reviewer #3 (Remarks to the Author):

In the revision the authors have all my remarks and especially did a great job on image analysis and providing more ultrastructural features of the Pitpna1 KO beta cells.

In general their work is very thorough and provides novel insights into beta cell dysfunction.

Reviewer #4 (Remarks to the Author):

Overall, the paper has much improved and data. especially on T2D islets are impressive!

I am still not convinced on the TUNEL/Ki67 stainings:

1) Pics in 3a show mostly T+ hormone- cells.

There is a large fraction of T+, which doesn't even colocalize with DAPI.

And the enlarged part shows random dots, not nuclear, mostly not beta-cells, two arrows point to cells which look like glucagon+, one arrow is hormone-, there is no T+insulin+, as reported in the quantification.

Why is DAPI excluded in the magnification? Please add.

Authors really seem to have problems with this staining.

Maybe you have some westerns left to blot for Caspase 3? Or stain for Caspase 3, as the TUNEL again has'nt worked?

The In Situ Cell Death Detection Kit (TUNEL) Roche kit usually works fine. There is no protocol given in the methods. Maybe proteinase K is needed?

Similar unusual TUNEL stains are presented in SupplFig2K. No DAPI colocalization, some random cytosolic dots. Not convincing.

2) Ki67 is equally confusing in SupplFig2K. Please change the graph to % ins+ cells, or maybe % is just missing.

How come that 50% of the acinar cells are in proliferation? This is a rare observation, and even many beta-cells proliferate here, although it's mouse, it's still quite high.

Usually, Ki67+ is around 0.5% (not 5%? as seen here) in mouse beta-cells under basal conditions (e.g. Cell Metabolism 23, 194–205, 2016; Nat Med. 2015 Apr;21(4):383-8).

Please re-evaluate stainings again. Check more samples, whether those shown are really representative? Just to make sure not to have some weak stains included in the paper.

Reviewer #1 (Remarks to the Author):

The authors demonstrate that PTPNA dependent PtdIns-4-P synthesis has an expected role to integrate the function of membrane compartments in pancreatic beta cells that are important for GSIS and other related beta cell functions—including insulin granule biogenesis and maturation, proinsulin processing, and resistance to ER stress and progressive apoptosis. Importantly, the function of PTPNA is generally similar in MIN6 cells, mouse islets/beta cells, and human islets/beta cells. My general feeling is that the data and analysis are exhaustive and detailed, and I generally have no question about what has been assembled here; however, the results section is difficult to read owing to the excessive detail. It seems the section could be improved considerably with some careful editing.

We thank Reviewer 1 for their supportive comments and the Results section has been edited to remove extraneous wording and to improve its clarity.

The effects of over expression or knock down of PTPNA are sometime small. For example, although the entire curve is mildly reduced, first phase insulin secretion is still intact in beta cell conditional KO mice (See Fig 2c).

We agree with Reviewer 1's assessment that loss of PTPNA in beta-cells (as depicted in experiments implementing the conditional beta-cell-specific knockout mouse or in treated isolated islets) does not completely abolish glucose-stimulated insulin secretion (GSIS). We would however highlight that the observed difference in GSIS in *Ins-Cre,Pitpna^{fllox/fllox}* mice is measured under an acute time frame (15 minutes), and that more importantly, *Ins-Cre,Pitpna^{fllox/fllox}* mice are hyperglycemic under random-fed steady-state conditions reflecting that under chronic conditions, PTPNA is a physiologically-relevant gene in beta-cell function. **I don't fully understand how the time interval of analysis (15 mn) can help explain the small effect on insulin secretion. The 15min interval is needed to reveal the first and 2nd phase secretion. Further, invoking hyperglycemia doesn't help the explanation.**

This seems unexpected if PTPNA is so central to beta cell function. I wonder how islets continue to function after PTPNA knockout? How do the islets get around the absence of such an important enzyme, unless other unknown mechanisms also play important roles?

The precise reason for the absence of a more severe phenotype is unknown to us at this time. We agree that "other unknown mechanisms also play important roles". One potential factor is a related family member, PTPNB, also mediates PtdIns-4-P-dependent recruitment of GOLPH3 to Golgi membranes and has previously been shown to act in redundant fashion to regulate development of the neocortex (Xie et al Dev Cell 2018). **Is this alternative isoform expressed in beta cells? At what level?** We believe that beta-cells (at least in the short term) can accommodate loss of PTPNA due to this built-in redundancy and we believe that many critical pathways have evolved to incorporate functional redundancy/overlap to avert a complete loss of function (i.e. cessation of insulin release) that leads to disease. **Stating that many pathways have evolved to circumvent various genetic mutations doesn't help explain anything in this instance. This is general assumption often used as an explanation.** However, based on the hyperglycemia observed in *Ins-Cre,Pitpna^{fllox/fllox}* mice, we hypothesize that redundant pathways cannot fully compensate for the chronic absence of PTPNA in beta-cells ultimately leading to beta-cell failure and hyperglycemia. **The important question is whether PtdIns-4-P production is central to the generation and secretion of insulin vesicles. I'm not sure whether the authors discuss this point somewhere in the manuscript. If not, they might consider adding a version of this text in the discussion.**

The analysis of PTPNA in human islets focuses our understanding of possible defects underlying T2D, which provide important translational results. The story is certainly written in a way to emphasize the critical importance of PTPNA in T2D. Regardless, it is difficult to know how PTPNA dysregulation contributes to the progression of T2D, especially whether it is the driving mechanism. Genetic/viral strategies nicely show that increased PTPNA improves GSIS in human islets/beta cells while decreased PTPNA impairs function. Regardless, the mechanism of PTPNA failure in T2D beta cells is unclear. This is an important weakness. Even with all the data, it is impossible to establish whether the loss of PTPNA is a cause of beta cell failure or one of the consequences of beta cell failure arising from the increased demand upon beta cells caused

by hepatic and peripheral insulin resistance. At least the authors show that increased PIPNA cause enhancement of beta cell function in islets from T2D patients.

We would respectfully disagree with the statement “it is impossible to establish whether the loss of PIPNA is a cause of beta cell failure or one of the consequences of beta cell failure”. In this study, we present our findings showing 1) beta-cell specific knockout mouse of *Pitpna* results in impaired GSIS, reduced beta-cell mass and random-fed hyperglycemia and 2) PIPNA is significantly reduced in T2D human islets. Together we would argue these observations point to a degree of causality in human beta-cell failure.

To be clear, we believe that beta-cell failure is likely not a singular event or caused by a single factor (DNA, RNA or protein). This is based on the published work showing reduced beta-cell expression of numerous genes in T2D human islets and we would argue it is virtually impossible with current technologies to lay claim to identifying ‘the sole cause for human beta-cell failure’. There may be hundreds of functionally-relevant genes down-regulated in a manner similar to PIPNA, and therefore, how likely is it that one factor preceded all of the other ‘consequences’? As discussed above, compensatory mechanisms enable beta-cells to respond to increased demand for insulin and after years of chronic duress, beta-cell failure ensues. Given that the full repertoire of functionally-relevant genes in the beta-cell is currently not fully known, as well as which of those genes are perturbed during human T2D is also not known, we acknowledge that diminished PIPNA is not likely the sole cause of beta-cell failure, but also that a ‘sole cause’ may not even exist and that it is not possible to ascertain with current technology. As Reviewer 1 notes, importantly, we show after restoring PIPNA expression in T2D islets to normal levels, we demonstrate that 1) glucose-stimulated insulin secretion is rescued in addition to insulin granule maturation and 2) expression of many ER stress proteins (i.e. CHOP, BIP, and PDI) is reversed.

While this is not evidence for loss of PIPNA as a ‘sole cause of beta-cell failure’, this demonstrates that restoration of PIPNA is adequate to restore GSIS in T2D beta-cells and that loss of PIPNA expression contributes to reducing the secretory output of beta-cells. **I agree with the authors that many defects might contribute to human beta cell failure in T2D, and that loss of PIPNA might be only one of many. Specifically focusing on human islets from T2Ds, can the authors gauge the relative contribution of PIPNA verses the many other genes that might contribute to beta cell failure. Specifically, is PIPNA a minor player that if dysregulated alone would be relatively benign?**

A mechanism of PIPNA failure in beta cells of T2D is conspicuously missing. This is disappointing. We are led to think it might be through dysregulated expression of miR-375 that suppresses PIPNA along with other genes. But the evidence is weak or absent in this regard. So, what is the mechanism of dysregulated PIPNA in human islets. Unfortunately, this important question is left open.

To address whether suppression of PIPNA in T2D islets is mediated by the microRNA pathway (i.e. miR-375), we performed western blotting on ARGONAUTE2 (AGO2), an RNA-binding protein and key mediator of microRNA-induced gene silencing including miR-375 (Tattikota et al. Cell Metabolism, 2014). Interestingly, similar to PIPNA, expression of AGO2 is dramatically reduced in T2D human islets (**new Figure 1i**). In our previous work, we showed beta-cell specific deletion of *Ago2* in mice impaired compensatory beta-cell proliferation and this indicated that microRNA-mediated gene silencing contributed to the adaptive proliferation of beta-cells in response to increased insulin demand. From this, we hypothesized that multiple microRNAs expressed in beta-cells including miR-375 suppress the secretory machinery to accommodate the cellular energy requirements for proliferation. Therefore, our new observation showing loss of AGO2 expression in T2D human islets may reflect the diminished capacity for compensatory proliferation in beta-cells during T2D. Furthermore, alluding to the discussion above, we tested whether loss of PIPNA in human islets affects AGO2 expression. We measured AGO2 expression by western blotting after inhibition of *PIPNA* in human islets and observed reduced AGO2 protein levels (**new Extended Data Figure 1h**). Based on this result, we hypothesize that loss of AGO2 is a consequence of loss of PIPNA in human islets, as loss of *Ago2* in beta-cells of mice (*Ins-Cre, Pitpna^{flox/flox}* mice) was shown to prompt increased expression of genes contributing to secretion including *Pitpna* (Extended Data Figure 1e). The observation that loss of PIPNA coincides with loss of AGO2 in T2D human islets indicates the silencing AGO2 is inadequate to restore expression of PIPNA and can also reflect the diminished capacity for increasing secretion and adaptive beta-cell proliferation to meet the increased insulin demand.

Lastly, a previous study has performed miRNA profiling using human T2D islets and noted differential expression of only 2 miRNAs (miR-187 and miR-345) (Locke et al, 2014 Diabetologia). Therefore, from both our own and published observations, reduced PITPNA expression in T2D human islets does not appear to result from increased activity of miRNAs including miR-375. **If not already done, perhaps the authors might consider adding this analysis into the discussion of the revised manuscript?**

In summary, we acknowledge the precise mechanism of reduced PITPNA during T2D remains unknown to us and while this is a limitation of our current study, we now show that loss of PITPNA in T2D human islets coincides with loss of AGO2 expression, the RNA-binding protein we previously demonstrated to mediate compensatory beta-cell proliferation. We also demonstrate that inhibition of PITPNA in human islets reduces AGO2 expression indicating loss of AGO2 follows loss of PITPNA. These results may suggest that loss of compensatory beta-cell proliferation is a consequence of reduced granule formation that characterizes loss of PITPNA and together these events incur the ER stress that is associated with beta-cell failure during T2D.

RESPONSE TO REVIEWERS

Reviewer #1:

I don't fully understand how the time interval of analysis (15 mn) can help explain the small effect on insulin secretion. The 15min interval is needed to reveal the first and 2nd phase secretion. Further, invoking hyperglycemia doesn't help the explanation.

To amend and clarify our previous response, throughout our study, we performed multiple experiments to assess the impact of *Pitpna*-deficiency on beta-cell function including: 1) insulin output in response to glucose challenge in *Ins-Cre, Pitpna^{flox/flox}* mice (shown in **Figure 2c**), 2) GSIS in isolated mouse and human islets (**Figures 2e and 4b**), and 3) by TIRF analysis in human islets (**Figures 4g and 4h**). Importantly, loss of *Pitpna* resulted in impaired exocytosis in all of these experiments and reflects the robustness of the effect of *Pitpna*-deficiency on beta-cell function. In addition, *Pitpna*-deficiency reduces beta-cell mass (**Figure 2l**). We believe these cumulative effects contribute to the metabolic dysfunction observed in *Ins-Cre, Pitpna^{flox/flox}* mice.

Our previous point regarding the time interval of analysis alludes to possible differences between experimental results from 'acute' treatments (i.e. 15-minute observational window following a supra-physiologic glucose treatment, or cumulative analysis over minutes and seconds in isolated mouse and human islets and during TIRF studies, respectively) versus those that reflect 'chronic' conditions. Most important is that conditional deletion of *Pitpna* in beta-cells of mice caused hyperglycemia under random-fed conditions. Taken together, it is not perfectly clear why *Pitpna*-deficiency does not completely abolish GSIS; however, whether measured over seconds or minutes, we consistently observed reduced GSIS as well as provide in vivo, physiologic evidence of this impairment (i.e. hyperglycemia).

Is this alternative isoform expressed in beta cells? At what level?

We assessed *PITPNB* expression in the single cell transcriptomic analysis from non-diabetic and T2D human islets (shown in new **Extended Data Figure 1h**). This result revealed *PITPNB* expression in beta-cells is not altered during T2D and that endogenous expression of *PITPNB* mRNA is comparable to *PITPNA* in this cell type.

Stating that many pathways have evolved to circumvent various genetic mutations doesn't help explain anything in this instance. This is general assumption often used as an explanation.

The conclusions we have drawn in this study are based on the experimental evidence we are presenting. We acknowledge this is a limitation of our current study.

The important question is whether PtdIns-4-P production is central to the generation and secretion of insulin vesicles. I'm not sure whether the authors discuss this point somewhere in the manuscript. If not, they might consider adding a version of this text in the discussion.

This aspect of the study is described in the Discussion starting at Lines 382-399. PtdIns-4-P contributes to mitochondrial and ER dynamics and this is discussed on Lines 412-417.

I agree with the authors that many defects might contribute to human beta cell failure in T2D, and that loss of *PITPNA* might be only one of many. Specifically focusing on human islets from T2Ds, can the authors gauge the relative contribution of *PITPNA* versus the many other genes that might contribute to beta cell failure. Specifically, is *PITPNA* a minor player that if dysregulated alone would be relatively benign?

We believe our study addresses precisely this point and that the answer to this question is: **No, *PITPNA* is not a minor player. Here we show dysregulation of *PITPNA* is a significant contributor of beta-cell failure in humans.** The evidence we present to support this is as follows: 1) beta-cell expression of *PITPNA* is significantly reduced in T2D human islets, 2) conditional deletion of *Pitpna* in beta-cells of mice results in hyperglycemia, 3) inhibition of *PITPNA* in human islets reduces GSIS, and 4) restoration of *PITPNA* in T2D human islets reverses beta-cell dysfunction. Together these results demonstrate that *PITPNA*-deficiency significantly contributes to beta-cell dysfunction in T2D human subjects.

In the context of other genes, as stated previously, we show that loss of *AGO2* expression is also reduced in human T2D islets (**Figure 1i**). We previously showed that conditional deletion of *Ago2* in beta-cells compromised compensatory proliferation of this cell type; however, in contrast to *Pitpna*-deficiency, loss of *Ago2* does not result in hyperglycemia (Tattikota et al 2013). We previously stated that inhibition of *PITPNA* in human islets reduces *AGO2* expression indicating that loss of *AGO2* in T2D islets is a consequence of reduced *PITPNA*. These results are consistent with the established model in humans that impaired beta-cell secretion precedes reduced beta-cell mass. Moreover, despite the absence of hyperglycemia in *Ago2*-deficient mice, we would not

call Ago2 a 'minor player' in beta-cell dysfunction. As mentioned previously, we would hypothesize that beta-cell failure is likely to involve hundreds or thousands of genes, and a stepwise loss of PITPNA and AGO2 is reflected in the loss of secretion and mass, respectively, and characterizes this process.

If not already done, perhaps the authors might consider adding this analysis into the discussion of the revised manuscript?

Possible dysregulation of miR-375 was discussed in Lines 444-448 in the Discussion section and we have now cited Locke et al Diabetologia 2014 as reference 88.

Reviewer #2:

The authors have adequately addressed my concerns. No further comments.

We thank Reviewer 2 for their supportive comments.

Reviewer #3:

In the revision the authors have all my remarks and especially did a great job on image analysis and providing more ultrastructural features of the *Pitpna1* KO beta cells. In general their work is very thorough and provides novel insights into beta cell dysfunction.

We thank Reviewer 3 for their supportive comments.

Reviewer #4:

Overall, the paper has much improved and data, especially on T2D islets are impressive!

We thank Reviewer 4 for their supportive comments.

I am still not convinced on the TUNEL/Ki67 stainings:

1) Pics in 3a show mostly T+ hormone- cells. There is a large fraction of T+, which doesn't even colocalize with DAPI. And the enlarged part shows random dots, not nuclear, mostly not beta-cells, two arrows point to cells which look like glucagon+, one arrow is hormone-, there is no T+insulin+, as reported in the quantification.

Why is DAPI excluded in the magnification? Please add. Authors really seem to have problems with this staining. Maybe you have some westerns left to blot for Caspase 3? Or stain for Caspase 3, as the TUNEL again has'nt worked? The In Situ Cell Death Detection Kit (TUNEL) Roche kit usually works fine. There is no protocol given in the methods. Maybe proteinase K is needed? Similar unusual TUNEL stains are presented in SupplFig2K. No DAPI colocalization, some random cytosolic dots. Not convincing.

Figure 3a and Extended Data Figure 2h have been revised with new TUNEL staining with the recommended kit from Roche (under Analytic Procedures on Line 519). As noted, colocalization of DAPI and TUNEL are shown in both *Pitpna*-deficient models.

2) Ki67 is equally confusing in SupplFig2K. Please change the graph to % ins+ cells, or maybe % is just missing. How come that 50% of the acinar cells are in proliferation? This is a rare observation, and even many beta-cells proliferate here, although it's mouse, it's still quite high. Usually, Ki67+ is around 0.5% (not 5%? as seen here) in mouse beta-cells under basal conditions (e.g. Cell Metabolism 23, 194–205, 2016; Nat Med. 2015 Apr;21(4):383-8). Please re-evaluate stainings again. Check more samples, whether those shown are really representative? Just to make sure not to have some weak stains included in the paper.

For clarification, *Pitpna* total knockout mice (as shown in **Extended Data Figure 2h-I**) die shortly after birth and both TUNEL and Ki67 staining were performed on pancreata isolated from mice at day P0 (not adult mice). This would be consistent with previous studies showing Ki-67 percentages in insulin+ cells during post-natal development at 5% or greater (Wang et al 2019 Diabetes).

REVIEWERS' COMMENTS

Reviewer #1 (Remarks to the Author):

Thanks for providing clear responses to my recent questions.

Reviewer #4 (Remarks to the Author):

The authors have well responded to the reviewers' questions. I have no more comments.
Congratulations to this comprehensive work!

RESPONSE TO REVIEWERS

Reviewer #1:

Thanks for providing clear responses to my recent questions.

We thank Reviewer 1 for their supportive comments.

Reviewer #2:

[No new comments received]

We thank Reviewer 2 for their previous supportive comments.

Reviewer #3:

[No new comments received]

We thank Reviewer 3 for their previous supportive comments.

Reviewer #4:

The authors have well responded to the reviewers' questions. I have no more comments. Congratulations to this comprehensive work!

We thank Reviewer 4 for their supportive comments.